# PANODIFFUSION: 360-DEGREE PANORAMA OUTPAINTING VIA DIFFUSION

**Tianhao Wu**[1][†]**, Chuanxia Zheng**[2] **& Tat-Jen Cham**[1]

[†]S-Lab, [1]Nanyang Technological University
`tianhao001@e.ntu.edu.sg, astjcham@ntu.edu.sg`
[2]University of Oxford
`cxzheng@robots.ox.ac.uk`

## ABSTRACT

Generating complete 360° panoramas from narrow field of view images is ongoing research as omnidirectional RGB data is not readily available. Existing GAN-based approaches face some barriers to achieving higher quality output, and have poor generalization performance over different mask types. In this paper, we present our 360° indoor RGB-D panorama outpainting model using latent diffusion models (LDM), called PanoDiffusion. We introduce a new bi-modal latent diffusion structure that utilizes both RGB and depth panoramic data during training, which works surprisingly well to outpaint *depth-free* RGB images during inference. We further propose a novel technique of introducing progressive camera rotations during each diffusion denoising step, which leads to substantial improvement in achieving panorama wraparound consistency. Results show that our PanoDiffusion not only significantly outperforms state-of-the-art methods on RGB-D panorama outpainting by producing diverse well-structured results for different types of masks, but can also synthesize high-quality depth panoramas to provide realistic 3D indoor models.

## 1 INTRODUCTION

Omnidirectional 360° panoramas serve as invaluable assets in various applications, such as lighting estimation (Gardner et al., 2017; 2019; Song & Funkhouser, 2019) and new scene synthesis (Somanath & Kurz, 2021) in the Augmented and Virtual Reality (AR & VR). However, an obvious limitation is that capturing, collecting, and restoring a dataset with 360° images is a *high-effort* and *high-cost* undertaking (Akimoto et al., 2019; 2022), while manually creating a 3D space from scratch can be a demanding task (Lee et al., 2017; Choi et al., 2015; Newcombe et al., 2011).

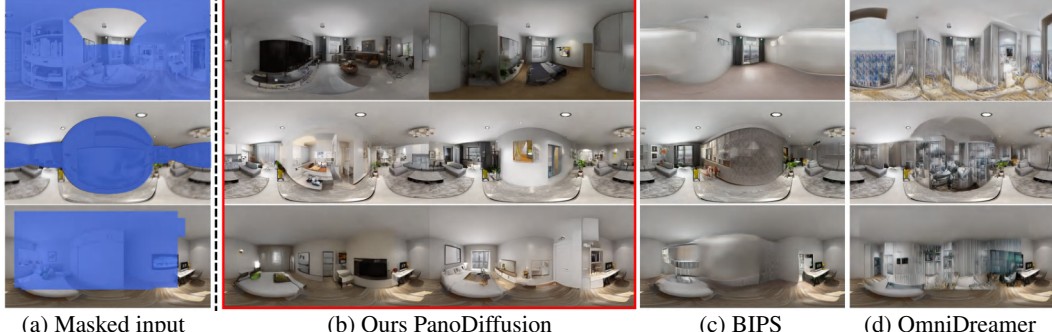

|  (a) Masked input | (b) Ours PanoDiffusion | (c) BIPS | (d) OmniDreamer |

Figure 1: **Example results of 360° Panorama Outpainting on various masks.** Compared to BIPS (Oh et al., 2022) and OmniDreamer (Akimoto et al., 2022), our model not only effectively generates semantically meaningful content and plausible appearances with many objects, such as beds, sofas and TV's, but also provides *multiple* and *diverse* solutions for this ill-posed problem. (Masked regions are shown in blue for better visualization. Zoom in to see the details.)

To mitigate this dataset issue, the latest learning methods (Akimoto et al., 2019; Somanath & Kurz, 2021; Akimoto et al., 2022; Oh et al., 2022) have been proposed, with a specific focus on *generating omnidirectional RGB panoramas from narrow field of view (NFoV) images*. These methods are typically built upon Generative Adversarial Networks (GANs) (Goodfellow et al., 2014), which have shown remarkable success in creating new content. However, GAN architectures face some notable problems, including **1)** mode collapse (seen in Fig. 1(c)), **2)** unstable training (Salimans et al., 2016), and **3)** difficulty in generating multiple structurally reasonable objects (Epstein et al., 2022). These limitations lead to obvious artifacts in synthesizing complex scenes (Fig. 1).

The recent endeavors of (Lugmayr et al., 2022; Li et al., 2022; Xie et al., 2023; Wang et al., 2023) directly adopt the latest latent diffusion models (LDMs) (Rombach et al., 2022) in image inpainting tasks, which achieve a stable training of generative models and spatially consistent images. However, specifically for a 360° panorama outpainting scenario, these inpainting works usually lead to grossly distorted results. This is because: **1)** the missing (masked) regions in 360° panorama outpainting is generally *much larger* than masks in normal inpainting and **2)** it necessitates generating *semantically reasonable objects* within a given scene, as opposed to merely filling in generic background textures in an empty room (as shown in Fig. 1 (c)). To achieve this, we propose an alternative method for 360° indoor panorama outpainting via the latest latent diffusion models (LDMs) (Rombach et al., 2022), termed as PanoDiffusion. Unlike existing diffusion-based inpainting methods, we introduce *depth* information through a novel *bi-modal* latent diffusion structure during the *training*, which is also significantly different from the latest concurrent works (Tang et al., 2023; Lu et al., 2023) that aims for *text-guided* 360° panorama image generation. Our *key motivation* for doing so is that the depth information is crucial for helping the network understand the physical structure of objects and the layout of the scene (Ren et al., 2012). It is worth noting that our model only uses partially visible RGB images as input during *inference*, *without* requirement for any depth information, yet achieving significant improvement on both RGB and depth synthesis (Tables 1 and 2).

Another distinctive challenge in this task stems from the unique characteristic of panorama images: **3)** both ends of the image must be aligned to ensure the integrity and *wraparound consistency* of the entire space, given that the indoor space lacks a definitive starting and ending point. To enhance this property in the generated results, we introduce two strategies: First, during the *training* process, a *camera-rotation* approach is employed to randomly crop and stitch the images for data augmentation. It encourages the networks to capture information from different views in a 360° panorama. Second, a *two-end alignment* mechanism is applied at each step of the *inference* denoising process (Fig. 4), which explicitly enforces the two ends of an image to be wraparound-consistent.

We evaluate the proposed method on the Structured3D dataset (Zheng et al., 2020). Experimental results demonstrate that our PanoDiffusion not only significantly outperforms previous state-of-the-art methods, but is also able to provide *multiple* and *diverse* well-structured results for different types of masks (Fig. 1). In summary, our main contributions are as follows:

- A new bi-modal latent diffusion structure that utilizes both RGB and depth panoramic data to better learn spatial layouts and patterns during training, but works surprisingly well to outpaint normal RGB-D panoramas during inference, *even without depth input*;

- A novel technique of introducing progressive camera rotations during *each* diffusion denoising step, which leads to substantial improvement in achieving panorama wraparound consistency;

- With either partially or fully visible RGB inputs, our PanoDiffusion can synthesize high-quality indoor RGB-D panoramas simultaneously to provide realistic 3D indoor models.

## 2 RELATED WORK

### 2.1 IMAGE INPAINTING/OUTPAINTING

Driven by advances in various generative models, such as VAEs (Kingma & Welling, 2014) and GANs (Goodfellow et al., 2014), a series of learning-based approaches (Pathak et al., 2016; Iizuka et al., 2017; Yu et al., 2018; Zheng et al., 2019; Zhao et al., 2020; Wan et al., 2021; Zheng et al., 2022) have been proposed to generate semantically meaningful content from a partially visible image. More recently, state-of-the-art methods (Lugmayr et al., 2022; Li et al., 2022; Xie et al., 2023;

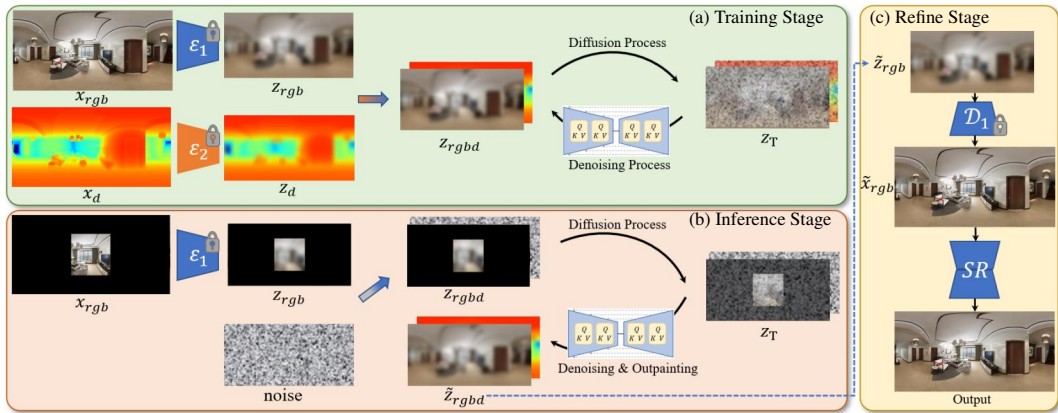

Figure 2: **The overall pipeline of our proposed PanoDiffusion method.** (a) During training, the model is optimized for RGB-D panorama synthesis, without the mask. (b) During inference, however, the depth information is *no longer needed* for masked panorama outpainting. (c) Finally, a super-resolution model is implemented to further enhance the high-resolution outpainting. We only show the input/output of each stage and omit the details of circular shift and adding noise. Note that the VQ-based encoder-decoders are pre-trained in advance, and fixed in the rest of our framework.

Wang et al., 2023) directly adopt the popular diffusion models (Rombach et al., 2022) for image inpainting, achieving high-quality completed images with consistent structure and diverse content. However, these diffusion-based models either focus on background inpainting, or require input text as guidance to produce plausible objects within the missing regions. This points to an existing gap in achieving comprehensive and contextually rich inpainting/outpainting results across a wider spectrum of scenarios, especially in the large scale 360° Panorama scenes.

## 2.2 360° PANORAMA OUTPAINTING

Unlike NFoV images, 360° panorama images are subjected to nonlinear perspective distortion, such as *equirectangular projection*. Consequently, objects and layouts within these images appear substantially distorted, particularly those closer to the top and bottom poles. The image completion has to not only preserve the distorted structure but also ensure visual plausibility, with the additional requirement of *wraparound-consistency at both ends*. Previous endeavors (Akimoto et al., 2019; Somanath & Kurz, 2021) mainly focused on deterministic completion of 360° RGB images, with BIPS (Oh et al., 2022) further extending this to RGB-D panorama synthesis. In order to generate diverse results (Zheng et al., 2019; 2021), various strategies have been employed. For instance, SIG-SS (Hara et al., 2021) leverages a symmetry-informed CVAE, while OmniDreamer (Akimoto et al., 2022) employs transformer-based sampling. In contrast, our PanoDiffusion is built upon DDPM, wherein each reverse diffusion step inherently introduces stochastic, naturally resulting in *multiple* and *diverse* results. Concurrently with our work, MVDiffusion (Tang et al., 2023) generates panorama images by sampling consistent multi-view images, and AOGNet (Lu et al., 2023) does 360° outpainting through an autoregressive process. Compared to the concurrent models, our PanoDiffusion excels in generating semantically multi-objects for large masked regions, *without the need of text prompts*. More importantly, PanoDiffusion is capable of simultaneously generating the corresponding RGB-D output, using only partially visible RGB images as input during the *inference*.

## 3 METHODS

Given a 360° image $x \in \mathbb{R}^{H \times W \times C}$, degraded by a number of missing pixels to become a masked image $x_m$, our main goal is to infer semantically meaningful content with reasonable geometry for the missing regions, while simultaneously generating visually realistic appearances. While this task is conceptually similar to conventional learning-based image inpainting, it presents greater challenges due to the following differences: **1)** our **output** is a *360° RGB-D panorama that requires wraparound consistency*; **2)** the **masked/missing** areas are generally *much larger* than the masks in traditional inpainting; **3)** our **goal** is to *generate multiple appropriate objects* within a scene, instead of simply filling in with the generic background; **4)** the completed results should be structurally plausible, which can be reflected by a reasonable depth map.

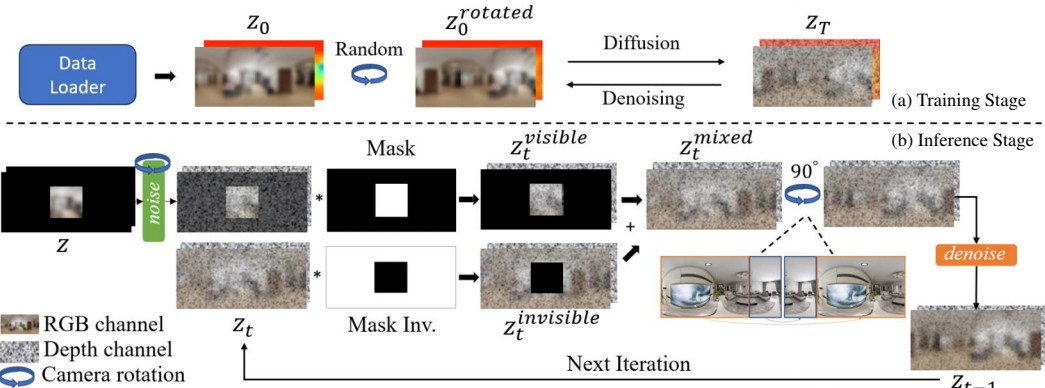

Figure 3: **Our LDM outpainting structure with camera rotation mechanism.** During training (a), we **randomly** select a rotation angle to generate a new panorama for data augmentation. During inference (b), we sample the visible region from the encoded features (above) and the invisible part from the denoising output (below). The depth map is *not needed*, and is set to random noise. At each denoising step, we crop a **90°**-equivalent area of the intermediate result from the right and stitch it to the left, denoted by the circle following $z_t^{mixed}$ — this strongly improves wraparound consistency.

To tackle these challenges, we propose a novel diffusion-based framework for 360° panoramic out-painting, called PanoDiffusion. The training process, as illustrated in Fig. 2(a), starts with two branches dedicated to RGB $x$ and depth $d_x$ information. Within each branch, following (Rombach et al., 2022), the input data is first embedded into the latent space using the corresponding pre-trained VQ model. These representations are then concatenated to yield $z_{rgbd}$, which subsequently undergoes the forward diffusion step to obtain $z_T$. The resulting $z_T$ is then subjected to inverse denoising, facilitated by a trained UNet, ultimately returning to the original latent domain. Finally, the pre-trained decoder is employed to rebuild the completed RGB-D results.

During inference, our system takes a masked RGB image as input and conducts panoramic outpainting. It is noteworthy that our proposed model does *not* inherently require harder-to-acquire depth maps as input, *relying solely on a partial RGB image* (Fig. 2(b)). The output is further super-resolved into the final image in a refinement stage (Fig. 2(c)).

### 3.1 PRELIMINARIES

**Latent Diffusion.** Our PanoDiffusion builds upon the latest Latent Diffusion Model (LDM) (Rombach et al., 2022), which executes the denoising process in the latent space of an autoencoder. This design choice yields a twofold advantage: it reduces computational costs while maintaining a high level of visual quality by storing the domain information in the encoder $\mathcal{E}(\cdot)$ and decoder $\mathcal{D}(\cdot)$. During the training, the given image $x_0$ is initially embedded to yield the latent representation $z_0 = \mathcal{E}(x_0)$, which is then perturbed by adding the noise in a Markovian manner:

$$q(z_t|z_{t-1}) = \mathcal{N}(z_t; \sqrt{1 - \beta_t}z_{t-1}, \beta_t I), \tag{1}$$

where $t = [1, \cdots, T]$ is the number of steps in the forward process. The hyperparameters $\beta_t$ denote the noise level at each step $t$. For the denoising process, the network in LDM is trained to predict the noise as proposed in DDPM (Ho et al., 2020), where the training objective can be expressed as:

$$\mathcal{L} = \mathbb{E}_{\mathcal{E}(x_0), \epsilon \sim \mathcal{N}(0,I), t}[||\epsilon_\theta(z_t, t) - \epsilon||_2^2] \tag{2}$$

**Diffusion Outpainting.** The existing pixel-level diffusion inpainting methods (Lugmayr et al., 2022; Horita et al., 2022) are conceptually similar to that used for image generation, except $x_t$ *incorporates partially visible information*, rather than purely sampling from a Gaussian distribution during the inference. In particular, let $x_0$ denote the original image in step 0, while $x_0^{visible} = m \odot x_0$ and $x_0^{invisible} = (1 - m) \odot x_0$ contain the visible and missing pixels, respectively. Then, as shown in Fig. 3, the reverse denoising sampling process unfolds as follows:

$$x_t^{visible} \sim q(x_t|x_{t-1}), \tag{3}$$

$$x_{t-1}^{invisible} \sim p_\theta(x_{t-1}|x_t), \tag{4}$$

$$x_{t-1} = m \odot x_{t-1}^{visible} + (1 - m) \odot x_{t-1}^{invisible}. \tag{5}$$

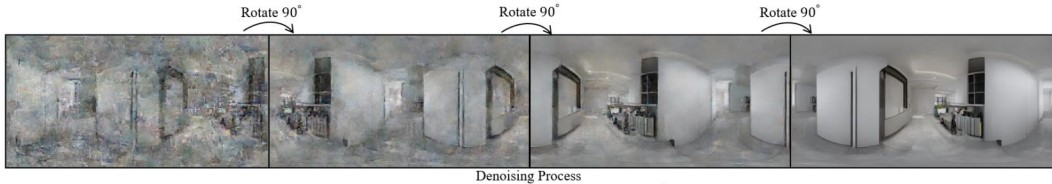

Figure 4: **An example of our two-end alignment mechanism.** During inference, we rotate the scene for 90° in *each* denoising step. Within a total of 200 sampling steps, our PanoDiffusion will effectively achieve wraparound consistency.

Here, $q$ is the forward distribution in the diffusion process and $p_\theta$ is the inverse distribution. After $T$ iterations, $x_0$ is restored to the original image space.

**Relation to Prior Work.** In contrast to these inpainting methods at pixel-level, our PanoDiffusion builds upon the LDM. Despite the fact that the original LDM provided the ability to inpainting images, such inpainting focuses on removing objects from the image, rather than generating a variety of meaningful objects in panoramic outpainting. In short, the $x_0$ is embedded into the latent space, yielding $z_0 = \mathcal{E}(x_0)$, while the subsequent sampling process follows the equations (3)-(5). The *key motivation* behind this is to perform our task on higher resolution 512×1024 panoramas. More importantly, we opt to go beyond RGB outpainting, and to deal with RGB-D synthesis (Sec. 3.3), which is useful for downstream tasks in 3D reconstruction. Additionally, existing approaches can *not* ensure the *wraparound consistency* during completion, while our proposed *rotational outpainting mechanism* in Sec. 3.2 significantly improves such a wraparound consistency.

## 3.2 WRAPAROUND CONSISTENCY MECHANISM

**Camera Rotated Data Augmentation.** It is expected that the two ends of any 360° panorama should be seamlessly aligned, creating a consistent transition from one end to the other. This is especially crucial in applications where a smooth visual experience is required, such as 3D reconstruction and rendering. To promote this property, we implement a *circular shift* data augmentation, termed *camera-rotation*, to train our PanoDiffusion. As shown in Fig. 3(a), we randomly select a rotation angle, which is subsequently employed to crop and reassemble image patches, generating a new panorama for training purposes.

**Two-Ends Alignment Sampling.** While the above *camera-rotation* technique can improve the model's implicit grasp of the wraparound consistency using the augmentation of data examples, it may *not* impose strong enough constraints on wraparound alignment of the results. Therefore, in the inference process, we introduce a *novel two-end alignment mechanism* that can be seamlessly integrated into our latent diffusion outpainting process. In particular, the reverse denoising process within the DDPM is characterized by multiple iterations, rather than a single step. During *each iteration*, we apply the camera-rotation operation, entailing 90° rotation of both the latent vectors and mask, before performing a denoising outpainting step. This procedure more effectively connects the two ends of the panorama from the previous step, resulting in significant improvement in visual results (as shown in Fig. 8). Without changing the size of the images, generating overlapping content, or introducing extra loss functions, we provide 'hints' to the model by rotating the panorama horizontally, thus enhancing the effect of alignment at both ends (examples shown in Fig. 4).

## 3.3 BI-MODAL LATENT DIFFUSION MODEL

In order to deal with RGB-D synthesis, one straightforward idea could be to use Depth as an explicit condition during training and inference, where the depth information may be compressed into latent space and then introduced into the denoising process of the RGB images via concatenation or cross-attention. However, we found that such an approach often leads to blurry results in our experiments (as shown in Fig. 11). Alternatively, using two parallel LDMs to reconstruct Depth and RGB images separately, together with a joint loss, may also appear to be an intuitive solution. Nonetheless, this idea presents significant implementation challenges due to the computational resources required for multiple LDMs.

Therefore, we devised a bi-modal latent diffusion structure to introduce depth information while generating high-quality RGB output. It is important to note that this depth information is *solely*

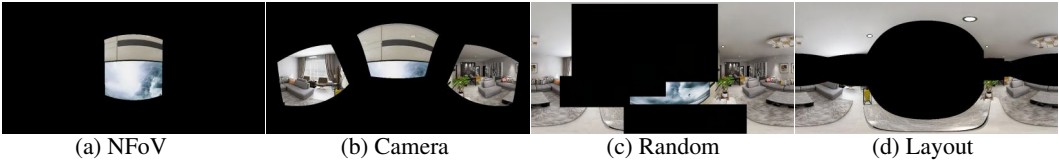

| (a) NFoV | (b) Camera | (c) Random | (d) Layout |

Figure 5: **Examples of various mask types.** See text for details.

*necessary during the training phase.* Specifically, we trained two VAE models independently for RGB and depth images, and then concatenate $z_{rgb} \in \mathbb{R}^{h \times w \times 3}$ with $z_{depth} \in \mathbb{R}^{h \times w \times 1}$ at the latent level to get $z_{rgbd} \in \mathbb{R}^{h \times w \times 4}$. The training of VAEs is exactly the same as in (Rombach et al., 2022) with downsampling factor $f$=4. Then we follow the standard process to train an unconditional DDPM with $z_{rgbd}$ via a variant of the original LDM loss:

$$\mathcal{L}_{RGB-D} = \mathbb{E}_{z_{rgbd}, \epsilon \sim \mathcal{N}(0,1), t}[\|\epsilon_\theta(z_t, t) - \epsilon\|_2^2], z_{rgbd} = \mathcal{E}_1(x) \oplus \mathcal{E}_2(d_x) \tag{6}$$

Reconstructed RGB-D images can be obtained by decoupling $z_{rgbd}$ and decoding. It is important to note that during training, we use the full RGB-D image as input, *without masks*. Conversely, during the inference stage, the model can perform outpainting of the masked RGB image directly *without any depth input*, with the fourth channel of $z_{rgbd}$ replaced by random noise.

### 3.4 REFINENET

Although mapping images to a smaller latent space via an autoencoder prior to diffusion can save training space and thus allow larger size inputs, the panorama size of $512 \times 1024$ is still a heavy burden for LDM (Rombach et al., 2022). Therefore, we adopt a two-stage approach to complete the outpainting task. Initially, the original input is downscaled to $256 \times 512$ as the input to the LDM. Correspondingly, the image size of the LDM output is also $256 \times 512$. Therefore, an additional module is needed to upscale the output image size to $512 \times 1024$. Since panorama images are distorted and the objects and layouts do not follow the regular image patterns, we trained a super-resolution GAN model for panoramas to produce visually plausible results at a higher resolution.

## 4 EXPERIMENTS

### 4.1 EXPERIMENTAL DETAILS

**Dataset.** We estimated our model on the Structured3D dataset (Zheng et al., 2020), which provides 360° indoor RGB-D data following equirectangular projection with a $512 \times 1024$ resolution. We split the dataset into 16930 train, 2116 validation, and 2117 test instances.

**Metrics.** For RGB outpainting, due to large masks, we should not require the completed image to be exactly the same as the original image, since there are many plausible solutions (e.g. new furniture and ornaments, and their placement). Therefore, we mainly report the following dataset-level metrics: 1) Fréchet Inception Distance (FID) (Heusel et al., 2017), 2) Spatial FID (sFID) (Nash et al., 2021), 3) density and coverage (Naeem et al., 2020). FID compares the distance between distributions of generated and original images in a deep feature domain, while sFID is a variant of FID that uses spatial features rather than the standard pooled features. Additionally, density reflects how accurate the generated data is to the real data stream, while coverage reflects how well the generated data generalizes the real data stream. For depth synthesis, we use RMSE, MAE, AbsREL, and Delta1.25 as implemented in (Cheng et al., 2018; Zheng et al., 2018), which are commonly used to measure the accuracy of depth estimates. Implementation details can be found in section 5.

**Mask Types.** Most works focused on generating omnidirectional images from NFoV images (Fig. 5(a)). However, partial observability may also occur due to sensor damage in 360° cameras. Such masks can be roughly simulated by randomly sampling a number of NFoV camera views within the panorama (Fig. 5(b)). We also experimented with other types of masks, such as randomly generated regular masks (Fig. 5(c)). Finally, the regions with floors and ceilings in panoramic images are often less interesting than the central regions. Hence, we also generated layout masks that muffle all areas except floors and ceilings, to more incisively test the model's generative power (Fig. 5(d)).

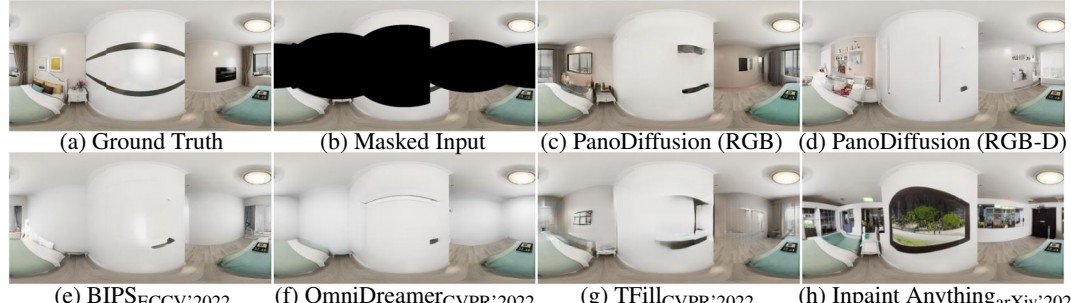

(a) Ground Truth  (b) Masked Input  (c) PanoDiffusion (RGB)  (d) PanoDiffusion (RGB-D)

(e) BIPS$_{ECCV'2022}$  (f) OmniDreamer$_{CVPR'2022}$  (g) TFill$_{CVPR'2022}$  (h) Inpaint Anything$_{arXiv'2023}$

Figure 6: **Qualitative comparison for RGB panorama outpainting.** Our PanoDiffusion generated more objects with appropriate layout, and with better visual quality. For BIPS and OmniDreamer, despite the seemingly reasonable results, the outpainted areas tend to fill the walls and lack diverse items. As for TFill, it generates blurry results for large invisible areas. For Inpaint anything, it generates multiple objects but they appear to be structurally and semantically implausible. Compared to them, PanoDiffusion generates more reasonable details in the masked region, such as pillows, paintings on the wall, windows, and views outside. More comparisons are provided in Appendix.

Table 1: **Quantitative results for RGB outpainting.** All models were re-trained and evaluated using the same standardized dataset. Note that, we tested all models *without* the depth input.

| Methods | Camera Mask | | | | NFoV Mask | | | | Layout Mask | | | | Random Box Mask | | | |
|---|---|---|---|---|---|---|---|---|---|---|---|---|---|---|---|---|
| | FID↓ | sFID↓ | D↑ | C↑ | FID↓ | sFID↓ | D↑ | C↑ | FID↓ | sFID↓ | D↑ | C↑ | FID↓ | sFID↓ | D↑ | C↑ |
| BIPS | 31.70 | 28.89 | 0.769 | 0.660 | 57.69 | 44.68 | 0.205 | 0.277 | 32.25 | 24.66 | 0.645 | 0.579 | 25.35 | 22.60 | 0.676 | 0.798 |
| OmniDreamer | 65.47 | 37.04 | 0.143 | 0.175 | 62.56 | 36.24 | 0.125 | 0.184 | 82.71 | 28.40 | 0.103 | 0.120 | 45.10 | 24.12 | 0.329 | 0.576 |
| LaMa | 115.92 | 107.69 | 0.034 | 0.082 | 125.77 | 136.32 | 0.002 | 0.006 | 129.77 | 35.23 | 0.018 | 0.043 | 45.25 | 24.21 | 0.429 | 0.701 |
| TFill | 83.84 | 61.40 | 0.075 | 0.086 | 93.62 | 76.13 | 0.037 | 0.027 | 97.99 | 43.40 | 0.046 | 0.052 | 46.84 | 30.72 | 0.368 | 0.574 |
| Inpainting Anything | 97.38 | 54.73 | 0.076 | 0.133 | 105.77 | 59.70 | 0.054 | 0.035 | 92.18 | 32.00 | 0.116 | 0.085 | 46.30 | 26.71 | 0.372 | 0.632 |
| RePaint | 82.84 | 84.39 | 0.096 | 0.105 | 95.38 | 82.35 | 0.0639 | 0.078 | 69.14 | 31.63 | 0.294 | 0.263 | 55.47 | 38.78 | 0.433 | 0.581 |
| PanoDiffusion | **21.55** | **26.95** | **0.867** | **0.708** | **21.41** | **27.80** | **0.790** | **0.669** | **23.06** | **22.39** | **1.000** | **0.737** | **16.13** | **20.39** | **1.000** | **0.883** |

**Baseline Models.** For RGB panorama outpainting, we mainly compared with the following state-of-the-art methods: including image inpainting models LaMa (Suvorov et al., 2022)$_{WACV'2022}$ and TFill (Zheng et al., 2022)$_{CVPR'2022}$, panorama outpainting models BIPS (Oh et al., 2022)$_{ECCV'2022}$ and OmniDreamer (Akimoto et al., 2022)$_{CVPR'2022}$, diffusion-based image inpainting models Repaint (Lugmayr et al., 2022)$_{CVPR'2022}$ and Inpaint Anything (Yu et al., 2023)$_{arXiv'2023}$. To evaluate the quality of depth panorama, we compare our method with three image-guided depth synthesis methods including BIPS (Oh et al., 2022), NLSPN (Park et al., 2020), and CSPN (Cheng et al., 2018). All models are retrained on the Structured3D dataset using their publicly available codes.

## 4.2 MAIN RESULTS

Following prior works, we report the quantitative results for RGB panorama outpainting with camera masks in Table 1. All instantiations of our model significantly outperform all state-of-the-art models. Specifically, the FID score is substantially better (relative 67.0% improvement).

It is imperative to note that our model is trained unconditionally, with masks only employed during the inference phase. Therefore, it is expected to *handle a broader spectrum of mask types*. To validate this assertion, we further evaluated our model with the baseline models across all four different mask types (displayed in Fig. 5). The results in Table 1 show that PanoDiffusion consistently outperforms the baseline models on all types of masks. Conversely, baseline models' performance displays significant variability in the type of mask used. Although the visible regions of the layout masks are always larger than the camera masks, the performances of baseline models on camera masks are significantly better. This is likely because the masks in the training process are closer to the NFoV distribution. In contrast, PanoDiffusion has a more robust performance, producing high-quality and diverse output images for all mask distributions.

The qualitative results are visualized in Fig. 6. Here we show an example of outpainting on a layout mask (more comparisons in Appendix). Besides the fact that PanoDiffusion generates more visually realistic results than baseline models, comparing the RGB (trained without depth) and RGB-D versions of our PanoDiffusion, in Fig. 6(c), some unrealistic structures are generated on the center

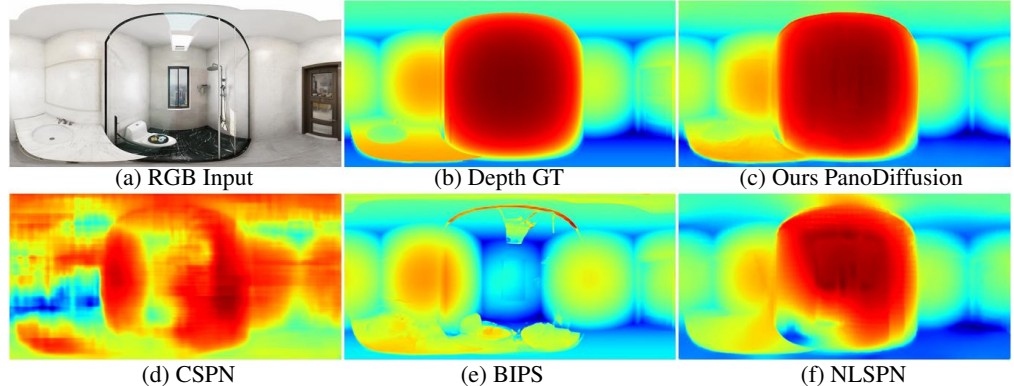

(a) RGB Input · (b) Depth GT · (c) Ours PanoDiffusion
(d) CSPN · (e) BIPS · (f) NLSPN

Figure 7: **Qualitative comparison for depth panorama synthesis.**

Table 2: **Depth map ablations**. All models are trained and evaluated on the Structured3D dataset.

| Noise Level | Camera Mask | | | | NFoV Mask | | | | Layout Mask | | | | Random Box Mask | | | |
|---|---|---|---|---|---|---|---|---|---|---|---|---|---|---|---|---|
| | FID↓ | sFID↓ | D↑ | C↑ | FID↓ | sFID↓ | D↑ | C↑ | FID↓ | sFID↓ | D↑ | C↑ | FID↓ | sFID↓ | D↑ | C↑ |
| no depth | 24.33 | 29.00 | 0.667 | 0.635 | 24.01 | 30.00 | 0.639 | 0.617 | 25.37 | 22.92 | 0.785 | 0.677 | 17.88 | 21.21 | 0.913 | 0.857 |
| 50% | 21.65 | 28.12 | 0.678 | 0.660 | 21.99 | 29.37 | 0.678 | 0.561 | 24.24 | 23.05 | 0.855 | 0.724 | 17.02 | 21.22 | 0.919 | 0.837 |
| 30% | 21.78 | 27.96 | 0.714 | 0.674 | 21.78 | 29.39 | 0.643 | 0.658 | 24.11 | 23.00 | 0.919 | 0.724 | 16.87 | 21.25 | 0.937 | 0.855 |
| 10% | 21.68 | 27.79 | 0.721 | 0.658 | 21.49 | 29.74 | 0.558 | 0.620 | 24.02 | 22.68 | 0.938 | 0.741 | 16.60 | 21.02 | 0.932 | 0.853 |
| full depth | 21.55 | 26.95 | 0.867 | 0.708 | 21.41 | 27.80 | 0.790 | 0.669 | 23.06 | 22.39 | 1.000 | 0.737 | 16.13 | 20.39 | 1.000 | 0.883 |

(a) **Usage of depth maps (training).** We use different sparsity levels of depth for training and the results (more intense color means better performance) verify the effectiveness of depth for RGB outpainting. It also proves that the model can accept sparse depth as input.

| Methods | Input Depth | FID↓ | sFID↓ | D↑ | C↑ |
|---|---|---|---|---|---|
| BIPS | fully visible | 29.74 | 30.59 | **0.931** | **0.721** |
| PanoDiffusion | | 21.90 | 26.78 | 0.829 | 0.693 |
| BIPS | partial visible | 31.70 | 28.89 | 0.769 | 0.660 |
| PanoDiffusion | | 22.34 | 26.74 | **0.856** | **0.686** |
| BIPS | fully masked | 68.79 | 42.62 | 0.306 | 0.412 |
| PanoDiffusion | | 21.55 | 26.95 | **0.867** | **0.708** |

(b) **Usage of depth maps (inference).** BIPS heavily relies on the availability of input depth during inference, while our model is minimally affected.

| Methods | Input Depth | RMSE↓ | MAE↓ | AbsREL↓ | Delta1.25↑ |
|---|---|---|---|---|---|
| BIPS | fully masked | 323 | 207 | 0.1842 | 0.8436 |
| CSPN | | 374 | 282 | 0.2273 | 0.6618 |
| NLSPN | | 284 | **183** | 0.1692 | 0.8544 |
| PanoDiffusion | | **276** | 193 | **0.1355** | **0.9060** |
| BIPS | partial visible | 247 | 136 | 0.1098 | 0.9032 |
| CSPN | | 291 | 195 | 0.1547 | 0.8182 |
| NLSPN | | 221 | 124 | **0.1058** | 0.9143 |
| PanoDiffusion | | **219** | **123** | 0.1127 | **0.9278** |

(c) **Depth panorama synthesis.** Our model outperforms baseline models in most of the metrics.

wall, and when we look closely at the curtains generated by the RGB model, the physical structure of the edges is not quite real. In contrast, the same region of RGB-D result (Fig. 6(d)) appears more structurally appropriate. Such improvement proves the advantages of jointly learning to synthesize depth data along with RGB images, *even when depth is not used during test time*, suggesting the depth information is significant for assisting the RGB completion.

## 4.3 ABLATION EXPERIMENTS

We ran a number of ablations to analyze the effectiveness of each core component in our PanoDiffusion. Results are shown in tables 2 and 3 and figs. 7 and 8 and discussed in detail next.

**Depth Maps.** In practice applications, depth data may exhibit sparsity due to the hardware limitations (Park et al., 2020). To ascertain the model's proficiency in accommodating sparse depth maps as input, we undertook a **training** process using depth maps with different degrees of sparsity (i.e., randomized depth value will be set to 0). The result is reported in Table 2(a). The denser colors in the table represent better performance. As the sparsity of the depth input decreases, the performance of RGB outpainting constantly improves. Even if we use 50% sparse depth for training, the result is overall better than the original LDM.

We then evaluated the importance of depth maps during **inference**, and compared it with the state-of-the-art BIPS (Oh et al., 2022), which is also trained with RGB-D images. The quantitative results are reported in Table 2(b). As can be seen, BIPS's performance appears to deteriorate significantly when

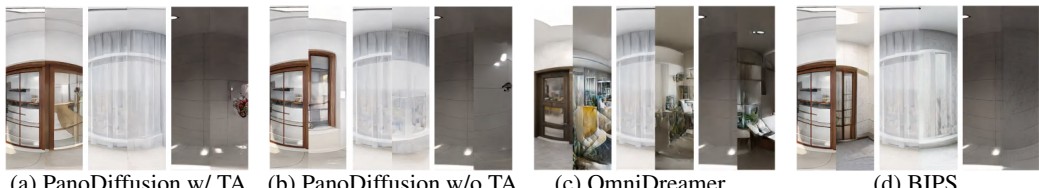

(a) PanoDiffusion w/ TA    (b) PanoDiffusion w/o TA    (c) OmniDreamer    (d) BIPS

Figure 8: **Examples of stitched ends of the outpainted images.** For each image, the left half was unmasked (i.e. ground truth), while the right half was masked and synthesized. The results generated with rotation are more naturally connected at both ends (a).

Table 3: **Two-end alignment ablations.** Using rotational outpainting, we achieve optimal consistency at both ends of the PanoDiffusion output.

| Methods \ Mask Type | Camera | NfoV | Layout | Random Box | Methods \ Mask Type | Camera | NfoV | Layout | Random Box |
|---|---|---|---|---|---|---|---|---|---|
| PanoDiffusion (w/ rotation) | **90.41** | **89.74** | **88.01** | **85.04** | PanoDiffusion (w/o rotation) | 125.82 | 128.33 | 128.10 | 128.19 |
| BIPS | 117.59 | 96.82 | 132.15 | 148.78 | OmniDreamer | 115.6 | 109.00 | 146.37 | 136.68 |
| LaMa | 119.51 | 119.39 | 133.54 | 136.35 | TFill | 155.16 | 157.60 | 136.94 | 122.96 |

the input depth visual area is reduced. Conversely, our PanoDiffusion is *not sensitive to these depth maps*, indicating that the generic model has successfully handled the modality. Interestingly, we noticed that having fully visible depth at test time did *not* improve the performance of PanoDiffusion, and in fact, the result deteriorated slightly. A reasonable explanation is that during the training process, the signal-to-noise ratios (SNR) of RGB and depth pixels are roughly the same within each iteration since no masks were used. However, during outpainting, the SNR balance will be disrupted when RGB input is masked and depth input is fully visible. Therefore, the results are degraded, but only slightly because PanoDiffusion has effectively learned the distribution of spatial visual patterns across all modalities, without being overly reliant on depth. This also explains why our model is more robust to depth inputs with different degrees of visibility.

Finally, we evaluated the depth synthesis ability of PanoDiffusion, seen in Table 2(c) and Fig. 7. The results show that our model achieves the best performance on most of the metrics and the qualitative results also show that PanoDiffusion is able to accurately estimate the depth map. This not only indicates that PanoDiffusion can be used for depth synthesis and estimation but also proves that it has learned the spatial patterns of panorama images.

**Two-end Alignment.** Currently, there is no metric to evaluate the performance of aligning the two ends of an image. To make a reasonable comparison, we make one side of the input image fully visible, and the other side fully masked and then compare the two ends of output. Based on the Left-Right Consistency Error (LRCE) (Shen et al., 2022) which is used to evaluate the consistency of two ends of the depth maps, we designed a new RGB-LRCE to calculate the difference between the two ends of the image: $LRCE = \frac{1}{N} \sum_{i=1}^{N} |P_{first}^{col} - P_{last}^{col}|$, and reported results in table 3.

The qualitative results are shown in fig. 8. To compare as many results, we only show the end regions that are stitched together to highlight the contrast. They show that the consistency of the two ends of the results is improved after the use of rotational outpainting, especially the texture of the walls and the alignment of the layout. Still, differences can be found with rotated outpainting. We believe it is mainly due to the fact that rotational denoising is based on the latent level, which may introduce extra errors during decoding.

## 5 CONCLUSION

In this paper, we show that our proposed method, the two-stage RGB-D PanoDiffusion, achieves state-of-the-art performance for indoor RGB-D panorama outpainting. The introduction of depth information via our bi-modal LDM structure significantly improves the performance of the model. Such improvement illustrates the effectiveness of using depth during training as an aid to guide RGB panorama generation. In addition, we show that the alignment mechanism we employ at each step of the denoising process of the diffusion model enhances the wraparound consistency of the results. With the use of these novel mechanisms, our two-stage structure is capable of generating high-quality RGB-D panoramas at 512×1024 resolution.

ACKNOWLEDGEMENT

This study is supported under the RIE2020 Industry Alignment Fund – Industry Collaboration Projects (IAF-ICP) Funding Initiative, as well as cash and in-kind contribution from the industry partner(s).

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

APPENDIX

The supplementary materials are organized as follows:

1. A video is added to illuminate the work with more results.

2. The reproducible code is included.

3. An additional PDF for implementation, training, metrics details, as well as more quantitative and qualitative results.

IMPLEMENT DETAILS

TRAINING OF VQ MODELS FOR RGB AND DEPTH PANORAMA

The reason why LDMs (Rombach et al., 2022) can be trained on larger image scales is that perceptual image compression is used so that the diffusion process can be conducted in latent space, with the decoder $\mathcal{D}$ used for returning the latent vector $z \in \mathbb{R}^{h \times w \times c}$ to high-resolution image $x \in \mathbb{R}^{H \times W \times C}$. During this process, the inherent spatial structure of the image does not change but is downscaled. Such spatial constancy is critical for our work as the partially visible regions will not be changed during outpainting, shown in Fig. 9.

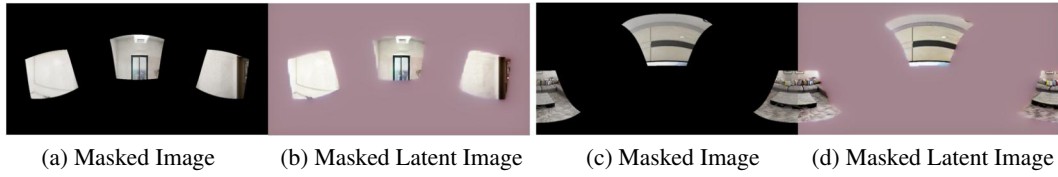

(a) Masked Image     (b) Masked Latent Image     (c) Masked Image     (d) Masked Latent Image

Figure 9: **Masking at different levels.** (a)(c) are pixel-level masked images, while (b)(d) are corresponding latent-level masked images. They are aligned well, except for the different mask values.

Two VQ models, $VQ_{rgb}$ and $VQ_{depth}$ with downsampling factor $f = 4$ are trained using RGB and depth data respectively. The output channel numbers of $VQ_{rgb}$ and $VQ_{depth}$ are set to 3 and 1. For the training of $VQ_{rgb}$, we finetuned on the pre-trained VQ-f4 model provided by Rombach et al. (Rombach et al., 2022). Due to the lack of a pre-trained model, $VQ_{depth}$ is trained from scratch. Both $VQ_{rgb}$ and $VQ_{depth}$ are trained for 30 epochs and we select the models that perform best on the validation data for the training of bi-modal LDM. Throughout the process, the datasets used during training, validation, and testing are exactly matched for $VQ_{rgb}$ and $VQ_{depth}$.

TRAINING OF LATENT DIFFUSION MODEL

The training of PanoDiffusion initially loaded the LSUN-Bedrooms (Yu et al., 2015) pre-trained model provided by official LDM (Rombach et al., 2022). Despite the fact that the LSUN-Bedrooms images are in normal view, where the objects and the layout are not subjected to equirectangular projection, we believe that it can provide useful priori knowledge of item semantics, texture, etc. for the outpainting of the indoor panoramas. At the same time, pre-trained $VQ_{rgb}$ and $VQ_{depth}$ are loaded and fixed during the training of our bi-modal LDM.

CAMERA-ROTATION DIRECTION

During the training stage, we restrict random angle rotation solely to the horizontal direction. As panorama images typically follow equirectangular projection, the distortion increases non-uniformly towards the top and bottom poles. Introducing random angle rotation in the vertical direction would lead to substantial changes in the projection results, and require more complex preprocessing, which will increase training costs. Conversely, distortion is uniform horizontally - image manipulation here only involves horizontal cropping and splicing, without the need for reprojection.

Here we show two examples where the camera has a 90-degree vertical rotation as Fig 10.

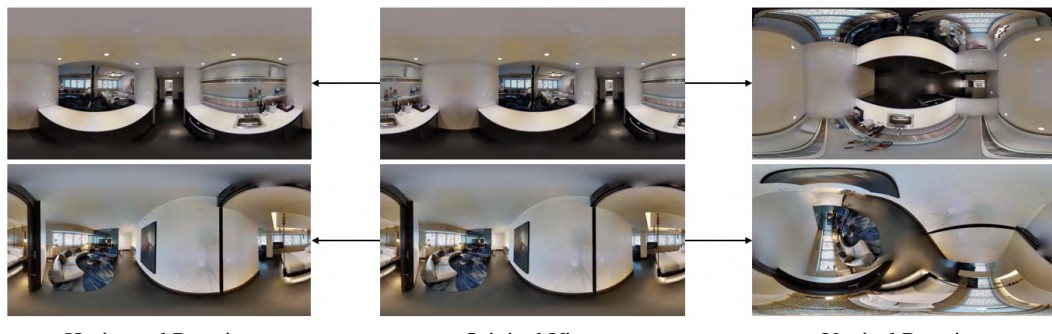

| Horizontal Rotation | Original View | Vertical Rotation |

Figure 10: **Examples of vertical and horizontal camera rotation.** No matter how many degrees the panorama is rotated horizontally, the distortion of each component remains unchanged. However, when the panorama is rotated vertically, even by a small angle, the distortion of the entire scene changes significantly, which can be a hindrance for PanoDiffusion to learn.

REFINENET IMPLEMENTATION

For the second stage of our structure, to generate visually plausible $512\times1024$ results, we trained a GAN for super-resolution. It consists of a generator G and a discriminator D. For training G, we use a weighted sum of the pixel-wise L1 loss and adversarial loss. The pixel-wise L1 loss is denoted as $L_{pixel}$, measuring the difference between the GT and the output panorama.

$$L_{pixel} = L_1(gt, G(gt_{lr})), \tag{7}$$

$$L_{adv} = \frac{1}{2}\mathbb{E}[(D(G(gt_{lr}) - 1)^2], \tag{8}$$

$$L_G = \lambda L_{pixel} + L_{adv}. \tag{9}$$

The training data is randomly downscaled from $512\times1024$ GT images to $128\times256$ or $256\times512$ and upscaled back to $512\times1024$ using the traditional interpolation method, which will erase details from GT images. Then they are used as the input of our super-resolution GAN, denoted as $gt_{lr}$. Here the value of $\lambda$ is set to 20 during the training.

QUANTITIVE METRICS

In this section, we will describe how the quantitative metrics used in this paper are implemented.

**Fréchet inception distance (FID)** FID (Heusel et al., 2017) is used to capture the similarity of generated images to real ones. We used the official PyTorch implementation of FID to evaluate the similarity between the final average pooling features of GT images and model outputs.

**Spatial FID (sFID)** Spatial FID (Nash et al., 2021) is a variant of FID, using spatial features rather than the standard pooled features. As standard pool_3 features compress spatial information to a large extent, making it less sensitive to spatial variability, mixed 6/conv features can provide a sense of spatial distributional similarity between models. sFID is calculated using the first 7 channels from the intermediate mixed 6/conv feature maps in order to obtain a feature space of size $7\times17\times17=2023$, which is comparable to the final average pooling features of size 2048.

**Density and Coverage** Density and coverage metrics are proposed by Naeem et al. (Naeem et al., 2020), who argue that even the latest version of the precision and recall metrics are still not reliable. Therefore, they proposed density and coverage, which can provide more interpretable and reliable signals. In this paper, we used their official implementation with nearest neighbor $k = 3$ to calculate the density and coverage of the final average pooling features of the GT panoramas and the generated output.

**Depth Estimation Metrics**  Given ground truth depth $D_{gt} = \{d_{gt}\}$ and predicted depth $D_{pred} = \{d_{pred}\}$, we use the following metrics to evaluate the quality of our depth estimates.

$$RMSE = \sqrt{\frac{1}{|D|} \sum ||d_{gt} - d_{pred}||^2}, \tag{10}$$

$$MAE = \frac{1}{|D|} \sum |d_{gt} - d_{pred}|, \tag{11}$$

$$AbsREL = \frac{1}{|D|} \sum \frac{|d_{gt} - d_{pred}|}{d_{gt}}, \tag{12}$$

$$Delta1.25 = \% \ of \ d_{pred} \ in \ D_{pred}, s.t. \ max(\frac{d_{pred}}{d_{gt}}, \frac{d_{gt}}{d_{pred}}) < 1.25 \tag{13}$$

ADDITIONAL RESULTS AND EXAMPLES

QUANTITATIVE COMPARISON FOR DEPTH-CONDITIONED LDM

As described in the main paper, to introduce depth information to aid RGB generation, an intuitive idea would be to use depth information as an explicit condition during training and inference. By compressing depth information into latent space $z_{depth}$, it can be introduced into the denoising process of the RGB images via cross-attention, denoted as depth-conditioned LDM (DC LDM). However, we have found that such an approach often leads to blurry results, as the image examples we have shown in the main paper (more examples are shown in Fig. 11).

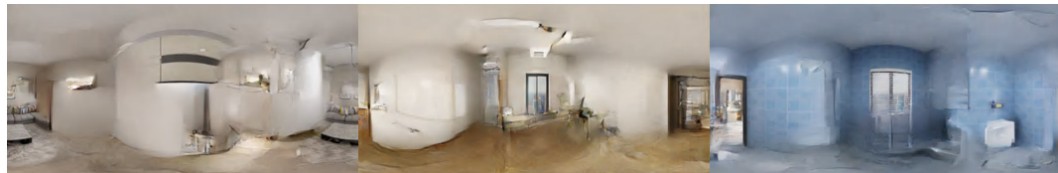

Figure 11: **Outpainting with a basic depth-conditioned LDM**. This leads to blurry results.

Since DC LDM is also trained with RGB-D images, here we report its performance on depth inputs with different degrees of visibility together with BIPS and our RGB-D PanoDiffusion, shown in Table 4.

Table 4: **Full quantitative results** for RGB outpainting with different depth input at test time.

| Methods | Input Depth | FID ↓ | sFID ↓ | Density ↑ | Coverage ↑ |
|---|---|---|---|---|---|
| BIPS | | 29.74 | 30.59 | **0.931** | **0.721** |
| DC LDM | fully visible | 77.75 | 44.47 | 0.051 | 0.086 |
| RGB-D PanoDiffusion | | **21.90** | **26.87** | 0.829 | 0.693 |
| BIPS | | 31.70 | 28.89 | 0.769 | 0.660 |
| DC LDM | partially visible | 77.77 | 44.44 | 0.054 | 0.080 |
| RGB-D PanoDiffusion | | **22.34** | **26.74** | **0.856** | **0.686** |
| BIPS | | 68.79 | 42.62 | 0.306 | 0.412 |
| DC LDM | fully masked | 78.15 | 44.29 | 0.048 | 0.073 |
| RGB-D PanoDiffusion | | **21.55** | **26.95** | **0.867** | **0.708** |

The results show that the DC LDM does not perform well in terms of both visual and quantitative results. Even if complete depth information is provided as a condition, the improvement in results is very marginal. This indicates that the conditional LDM structure cannot make good use of depth information to assist RGB panorama outpainting, which proves the effectiveness of our bi-modal LDM structure.

FULL QUANTITATIVE COMPARISON FOR REFINENET

As we only show the results of our RefineNet on camera and NFoV mask in the main paper, here we report the full quantitative results on all types of masks with both RGB and RGB-D PanoDiffusion

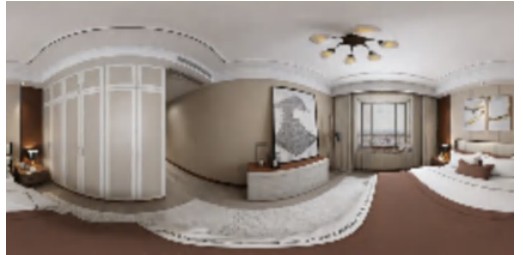 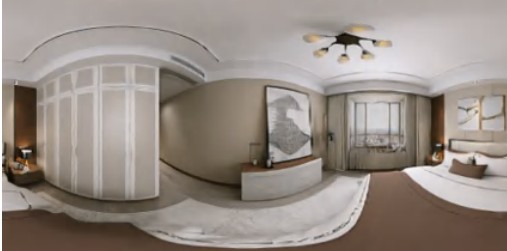

(a) Classical Interpolation Output          (b) Super Resolution Output

Figure 12: **Examples of upscaled results using classical interpolation method and out super-resolution GAN.**

(Table 5). A visualized comparison example is shown in Fig. 12. The results show that our super-resolution GAN improves the quality of PanoDiffusion output comprehensively, except for a slight degradation of the sFID of RGB PanoDiffusion on the NFoV mask.

Table 5: **Full quantitative results for RefineNet.** (-): classical interpolation. (+): super-resolution.

| Mask Type | Version | SR | FID ↓ | sFID ↓ | Density ↑ | Coverage ↑ | Mask Type | Version | SR | FID ↓ | sFID ↓ | Density ↑ | Coverage ↑ |
|---|---|---|---|---|---|---|---|---|---|---|---|---|---|
| Camera | RGB-D | - | 24.29 | 28.05 | 0.805 | 0.663 | Layout | RGB-D | - | 26.16 | 24.21 | 0.920 | 0.689 |
| | | + | **21.55** | **26.95** | **0.867** | **0.708** | | | + | **23.06** | **22.39** | **1.000** | **0.737** |
| | RGB | - | 26.75 | 29.89 | 0.581 | 0.570 | | RGB | - | 28.18 | 24.50 | 0.751 | 0.650 |
| | | + | **24.33** | **29.00** | **0.667** | **0.635** | | | + | **25.37** | **22.92** | **0.785** | **0.677** |
| NFoV | RGB-D | - | 23.96 | 28.19 | 0.775 | 0.645 | Random Box | RGB-D | - | 20.05 | 22.77 | 0.996 | 0.836 |
| | | + | **21.41** | **27.80** | **0.790** | **0.669** | | | + | **16.13** | **20.39** | **1.000** | **0.883** |
| | RGB | - | 26.72 | **29.94** | 0.648 | 0.595 | | RGB | - | 21.77 | 23.72 | 0.853 | 0.800 |
| | | + | **24.01** | 30.00 | **0.639** | **0.617** | | | + | **17.88** | **21.21** | **0.913** | **0.857** |

ABLATION STUDY ON CAMERA-ROTATION ANGLES

We additionally explored the effect of different rotation angles, including 180°, 90° (chosen for our final result), and 45°, on the outpainting results, seen as Table 6. The results show that the wrap-around consistency of outpainting results is improved across all settings. Compared to 180°, 90° leads to better consistency. However, diminishing the angle further to 45° did not lead to additional improvements. We believe this is reasonable, as the model is expected to generate coherent content when the two ends are in contact for enough denoising steps. Therefore, smaller rotation angles than 90° and longer connections do not necessarily lead to more consistent results.

Table 6: **Camera-rotation angles ablations.**

| Methods \ Mask Type | Camera | NfoV | Layout | Random Box | End |
|---|---|---|---|---|---|
| PanoDiffusion(w/o rotation) | 125.82 | 128.33 | 128.10 | 128.19 | 132.69 |
| PanoDiffusion(180°) | 95.11 | 96.57 | 90.93 | 85.23 | 119.60 |
| PanoDiffusion(90°) | 90.41 | 89.74 | 88.01 | 85.04 | 116.77 |
| PanoDiffusion(45°) | 90.67 | 90.25 | 87.65 | 86.50 | 112.47 |

ADDITIONAL VISUALIZATION EXAMPLES OF RGB PANORAMA OUTPAINTING

Due to page limitations, we only provide one group of comparative results for RGB outpainting in the main paper. Here we will provide more visualization examples, shown in Fig. 13. Same as in the main paper, we compare PanoDiffusion with LaMa (Suvorov et al., 2022), TFill (Zheng et al., 2022), OmniDreamer (Akimoto et al., 2022), BIPS (Oh et al., 2022), Repaint (Lugmayr et al., 2022), and Inpaint Anything (Yu et al., 2023) on different types of masks. It can be seen that our method outperforms the baseline models by generating various objects with appropriate layout, and with better visual quality. Besides, to prove that our PanoDiffusion can perform diverse and plausible completions on a given input, we provide two different outpainting results for each example.

ADDITIONAL VISUALIZATION EXAMPLES OF DEPTH PANORAMA SYNTHESIS

Due to page limitations, we only provide one group of comparative results for Depth synthesis in the main paper. Here we will provide more visualization examples, shown in Fig. 14. Same as in the main paper, we compare PanoDiffusion with BIPS (Oh et al., 2022), NLSPN (Park et al., 2020), and CSPN (Cheng et al., 2018). It can be seen that our method outperforms the baseline models by accurately estimating the depth map.

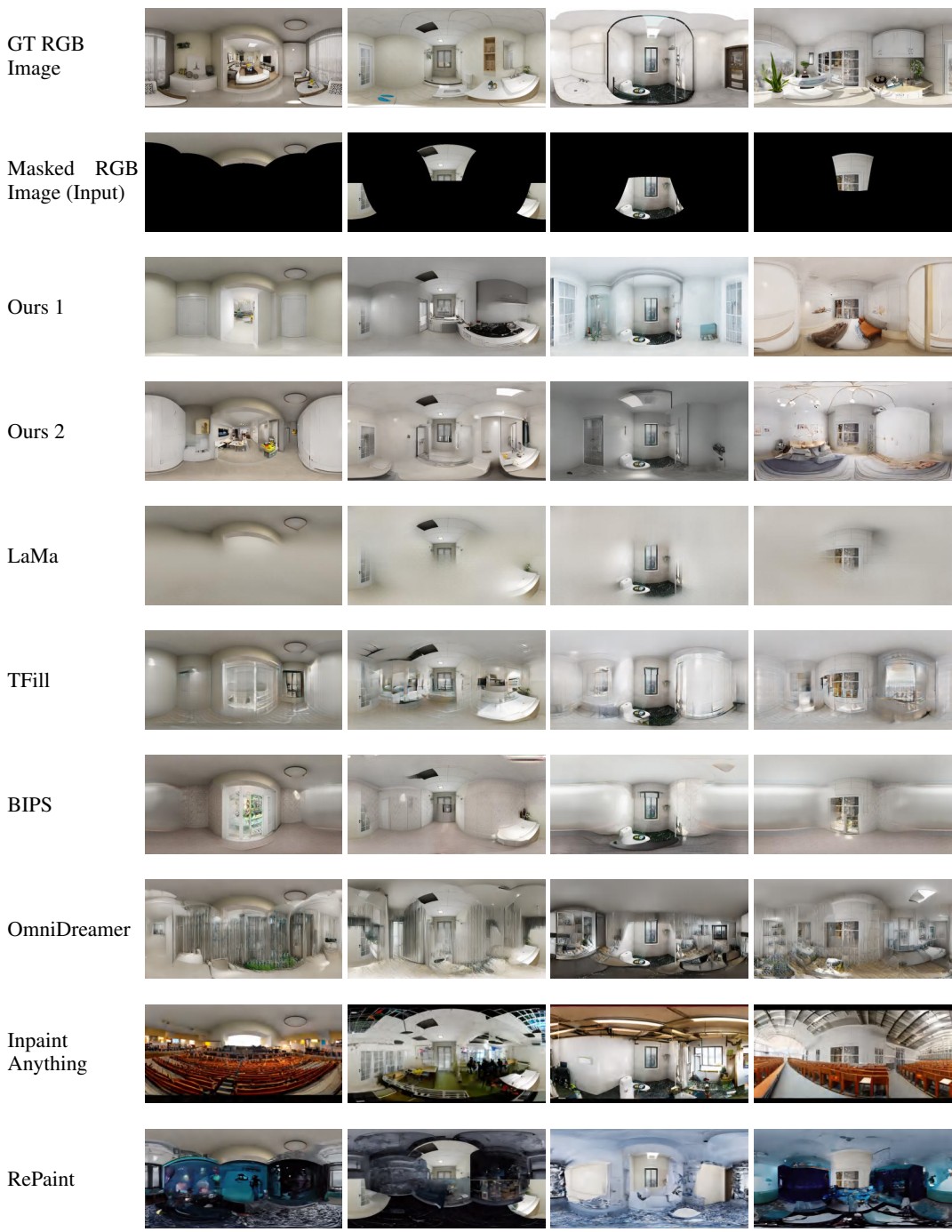

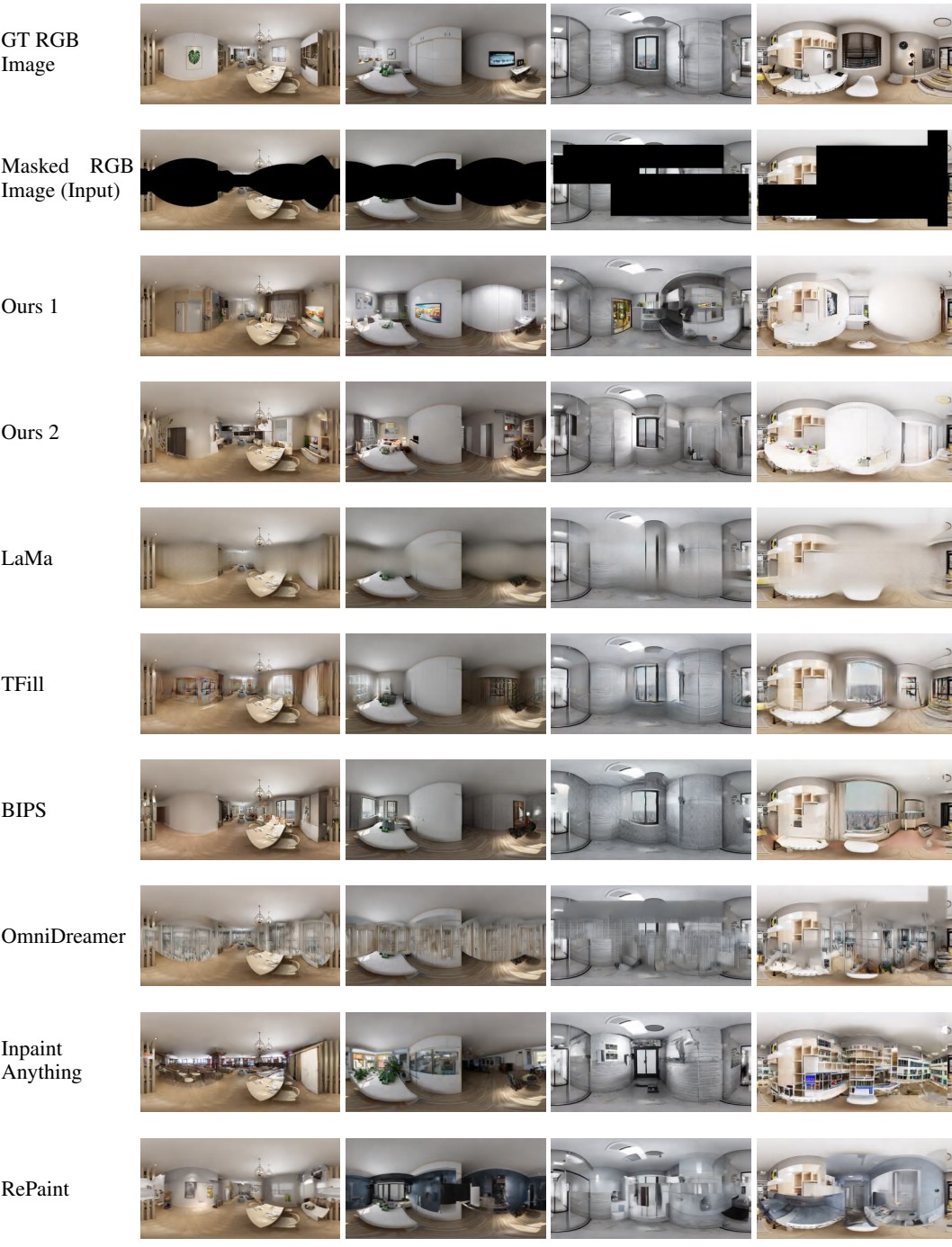

Figure 13: **Additional qualitative comparisons for RGB panorama outpainting.** Our PanoDiffusion generated more objects with appropriate layout, and with better visual quality. Please zoom in to see the details.

QUALITATIVE RESULTS OF ZERO-SHOT TEST ON MATTERPORT3D DATASET.

To test the generalization capability of PanoDiffusion, we conducted additional tests using a set of panorama images from the Matterport3D dataset. Here we provide six groups of examples from the outpainting results, which show that our model can have a decent outpainting effect on the real panorama dataset as well.

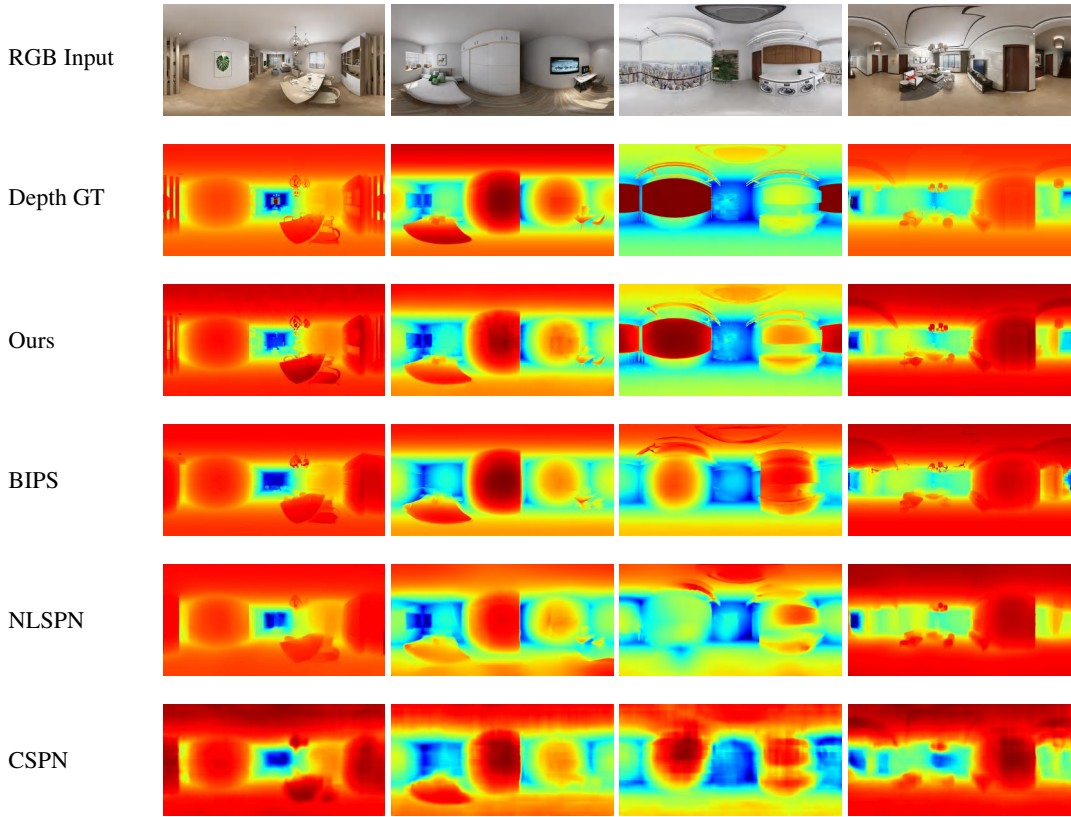

Figure 14: **Additional qualitative comparisons for Depth panorama synthesis.** Our PanoDiffusion achieves most accurate estimation. Please zoom in to see the details.

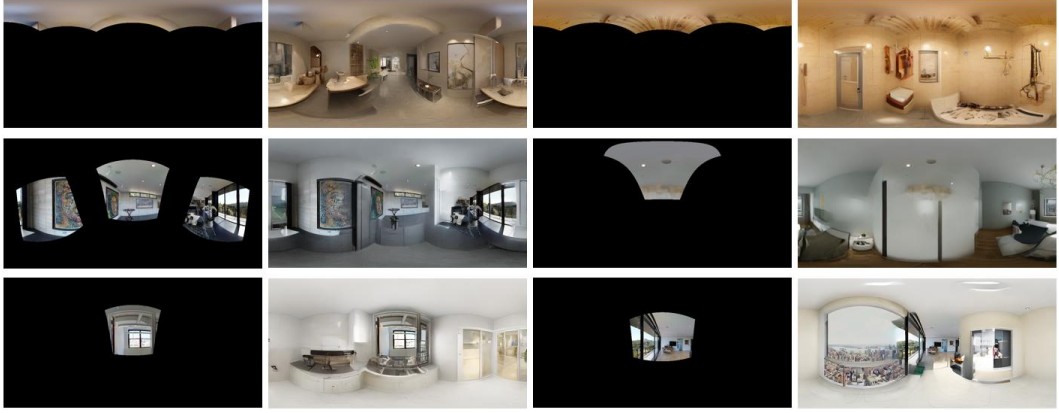

Figure 15: **Results of zero-shot test on Matterport3D dataset.** Zoom in to see the details.

QUALITATIVE RESULTS OF DISCRETE MASK ABLATION.

To explicitly assess our model's performance with discrete masks, we flipped the camera mask - swapping the originally visible and invisible parts to simulate this situation. This type of mask is equivalent to randomly sampling several NFoV masks and making them invisible. Here we provide examples from the outpainting results as Fig 16.

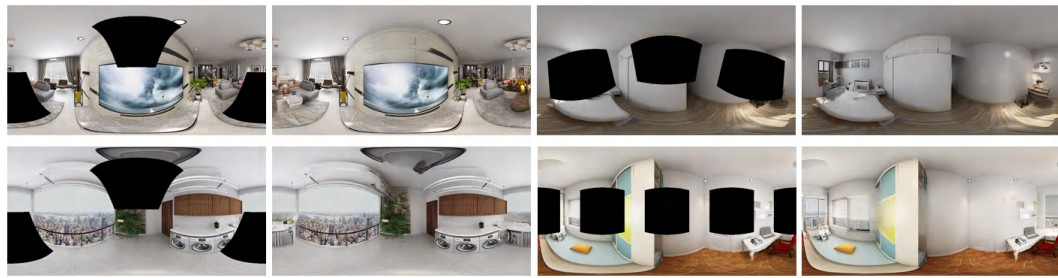

Figure 16: **Outpainting Results on discrete masks.**

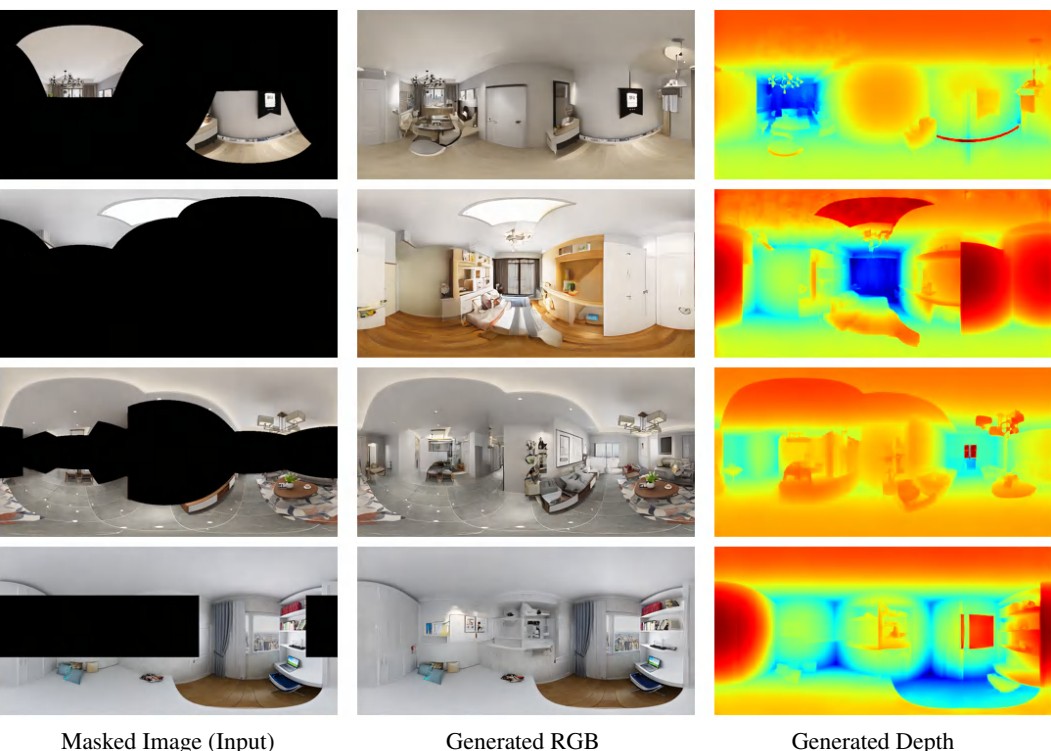

Masked Image (Input)    Generated RGB    Generated Depth

Figure 17: **Synthesized RGB-D Panorama Outpainting Results.**

QUALITATIVE RESULTS OF SYNTHESIZED RGB-D PANORAMA RESULTS.

In Fig. 17 we provide some synthesized RGB-D panorama examples where RGB is partially visible and depth is fully masked. The results show that our PanoDiffusion can outpainting plausible and consistent RGB-D panoramas simultaneously.

COMPLEXITY ANALYSIS

Table 7: **Training and inference time comparison**

| Method | Type | Depth | Training (mins/epoch) | Inference (sec/image) |
|---|---|---|---|---|
| PanoDiffusion | bi-modal LDM | + | 82 | 5 |
| BIPS | GAN | + | 131 | <1 |
| RePaint | Diffusion model | - | 78 | 45 |
| LDM | LDM | - | 72 | 4 |
| OmniDreamer | Transformer + VQGAN | - | 158 | 61 |

For training, we compared average training time (minutes) for one epoch of PanoDiffusion against baseline models on the same devices, using the same batch size 4 and the same training dataset. For inference, we compared the time (seconds) required to infer a single image.

The results show that while our model is not the fastest, it remains within a reasonable and acceptable range. It's also noteworthy that, compared to the original LDM framework, our bi-modal structure achieves a significant improvement in the quality of outpainting. This improvement comes without a proportionate increase in resource consumption – we observed only a modest increase of 13.8% in training time and 25% in inference time.

