# OpenReview forum: "PanoDiffusion: 360-degree Panorama Outpainting via Diffusion"
_ICLR.cc/2024/Conference — ICLR 2024 poster_

### Official Review · Reviewer_dEjQ · 2023-10-15

**Soundness:** 3 good
**Presentation:** 3 good
**Contribution:** 3 good
**Rating:** 6
**Confidence:** 3

**Summary:**

The paper presents a diffusion model for panoramic image generation. A two-stage RGB-D PanoDiffusion model is proposed for indoor RGB-D panorama outpainting. The model taks depth information as input and processed through a bi-modal LDM structure. As the results shown, the use of depth information enhances the generation of RGB panoramas, and the alignment mechanism ensures wraparound consistency in the results. The method can be used to generate RGB-D panoramas at 512×1024 resolution.

**Strengths:**

The RGB-D fusion and combination mechanism is brought to the field of panoramic image generation via using the latent diffusion models. A RGB-D panoramic outpainting model is proposed to perform indoor 360-degree image generation.

A bi-modal latent diffusion structure is proposed to combine RGB and depth information during training the diffusion model for panoramic image generation.

A camera-rotation method is proposed to perform a stronger data augmentation. The two-end method can be used to crop a 90-degree equivalent area and stitch to the opposite sides to perform additional data augmentation to improve the wraparound consistency.

**Weaknesses:**

As compared to the previous methods, the depth information is additionally added to train the diffusion model. Even though the depth information is not needed for inference.

The proposed RGB-D framework is constructed by using two parallel LDM to reconstruct the depth and RGB images separately. This structure might result in a larger and more complex model architecture than using a shared or depth-conditional LDM.

An additional module is needed to refine and upscale the low-resolution image output to a high-resolution image. However, a pre-trained super-resolution GAN model is needed to perform such a refinement.

**Questions:**

Apart from the visualization results, how about the evaluation results of using concatenation or cross-attention in the so-called depth-conditional diffusion model?

How is the cross-attention operation used in the depth-conditional method? Whether the authors try other advanced multimodal fusion methods to better combine the features from RGB and depth?

It would be better to ablate the camera-rotation data augmentation method.

In the depth panoramic synthesis, how about the comparison to some RGB-based depth estimation methods? For example, in the case of using fully masked input.

How about the runtime or complexity analysis of the proposed method, since the parallel LDM are used for RGB and depth separately.

---

> ### Author Response · Authors · 2023-11-20
> **Official Comment by Authors (1/2)**
>
> We appreciate the reviewer's valuable feedback and suggestions. We would like to provide a detailed response to each comment.
>
> > Q1: The proposed RGB-D framework is constructed by using two parallel LDM to reconstruct the depth and RGB images separately. This structure might result in a larger and more complex model architecture than using a shared or depth-conditional LDM.
>
> We do not use two parallel LDM's. As is claimed in Section 3.3, using two parallel LDMs to reconstruct Depth and RGB images separately will incur excessive computational resources. To avoid this, **our PanoDiffusion uses two VAE encoders, but only one LDM, with depth info as concatenated channels**. Therefore, the computational cost does not increase significantly compared to the original conditional LDM (see Q6).
>
> > Q2: Apart from the visualization results, how about the evaluation results of using concatenation or cross-attention in the so-called depth-conditional diffusion model?
>
> We've additionally tested depth-conditional diffusion models that use the usual concatenation and cross-attention conditioning on all types of masks. The results show our PanoDiffusion consistently outperforming the traditional depth-conditional LDM formulation.
>
> |                          | Camera    |           |           |           | NFoV       |           |           |           |
> | ------------------------ | --------- | --------- | --------- | --------- | ---------- | --------- | --------- | --------- |
> | Method                   | FID       | sFID      | D         | C         | FID        | sFID      | D         | C         |
> | PanoDiffusion            | **21.55** | **26.95** | **0.867** | **0.708** | **21.41**  | **27.80** | **0.790** | **0.669** |
> | DC-LDM (cross-attention) | 51.77     | 37.96     | 0.275     | 0.280     | 48.14      | 39.17     | 0.248     | 0.271     |
> | DC-LDM (concatenation)   | 114.59    | 51.96     | 0.023     | 0.018     | 112.54     | 64.80     | 0.046     | 0.006     |
> |                          | **Layout**   |           |           |           | **Random Box** |           |           |           |
> | Method                   | FID       | sFID      | D         | C         | FID        | sFID      | D         | C         |
> | PanoDiffusion            | **23.06** | **22.39** | **1.000** | **0.737** | **16.13**  | **20.39** | **1.000** | **0.883** |
> | DC-LDM (cross-attention) | 40.63     | 25.52     | 0.476     | 0.450     | 30.25      | 23.65     | 0.597     | 0.719     |
> | DC-LDM (concatenation)   | 105.79    | 32.90     | 0.060     | 0.054     | 64.49      | 30.38     | 0.237     | 0.471     |
>
>
>
> > Q3: How is the cross-attention operation used in the depth-conditional method? Whether the authors try other advanced multimodal fusion methods to better combine the features from RGB and depth?
>
> For depth-conditional LDM's, we pretrained a VQVAE for depth, and kept the encoder. The encoded depth is fed into the LDM as a condition in the usual two options, either via concatenation, or by cross-attention. Additionally, to improve robustness, we allowed the LDM to update the depth-condition encoder weights during training. However, these methods led to unsatisfactory outpainting results, often resulting in blurry content.
>
> We did not investigate more advanced multimodal fusion baselines, because our main task is RGB **outpainting without any depth input at inference time**.

---

> > ### Author Response · Authors · 2023-11-20
> > **Official Comment by Authors (2/2)**
> >
> > > Q4: It would be better to ablate the camera-rotation data augmentation method.
> >
> > We would like to emphasize that our main camera-rotation contribution, involving 90° rotations, is a process undertaken **during inference time**, and not a data augmentation method. For the ablation results for this inference-time process, please refer to Fig. 8 and Table 3 in the paper.
> >
> > Nonetheless, we had also carried out camera-rotation data augmentation during training, involving random rotations. To conduct an ablation for this part, we re-trained our model without camera-rotation data augmentation. The results indicate that our method with data augmentation is better than without augmentation, especially for Density and Coverage metrics. Besides, the outpainted results with this training augmentation have lower LRCE, i.e., better wrap-around consistency.
> >
> > |                  | Camera    |           |           |           |           | NFoV       |           |           |           |           |
> > | ---------------- | --------- | --------- | --------- | --------- | --------- | ---------- | --------- | --------- | --------- | --------- |
> > |                  | FID       | sFID      | D         | C         | LRCE      | FID        | sFID      | D         | C         | LRCE      |
> > | w/ augmentation  | **21.55** | **26.95** | **0.867** | **0.708** | **90.41** | **21.41**  | **27.80** | **0.790** | **0.669** | **89.74** |
> > | w/o augmentation | 21.93     | 27.56     | 0.813     | 0.706     | 98.92     | 22.33      | 29.19     | 0.680     | 0.642     | 90.39     |
> >
> >
> > > Q5: In the depth panoramic synthesis, how about the comparison to some RGB-based depth estimation methods?
> >
> > As suggested, we compared our PanoDiffusion with two RGB-based depth estimation methods, PanoFormer (ECCV'2022) [1] and ACDNet(AAAI'2022) [2], which are designed not only for depth estimation but specifically for panorama depth estimation.
> >
> > | **Methods**   | **RMSE ↓** | **MAE ↓** | **AbsREL ↓** | **Delta1.25 ↑** |
> > | ------------- | ---------- | --------- | ------------ | --------------- |
> > | ACDNet        | 353        | 209       | 0.2065       | 0.7804          |
> > | PanoFormer    | **268**    | **164**   | 0.2777       | 0.8863          |
> > | PanoDiffusion | 276        | 193       | **0.1355**   | **0.9060**      |
> >
> > PanoDiffusion performed better than Panoformer on AbsREL and Delta1.25, but poorer on RMSE and MAE. We believe that this is likely due to the primary objectives of each model. For PanoDiffusion, we are more concerned with the outpainting quality of the RGB panoramas, with depth as auxiliary information. **Accurate depth synthesis reflects that PanoDiffusion has adequately learned the information about depth as expected, while more importantly having the ability to conduct high quality RGB outpainting.** Conversely, ACDNet and PanoFormer are designed to prioritize the quality of depth output, with RGB images serving as the primary input.
> >
> > > Q6: How about the runtime or complexity analysis of the proposed method, since the parallel LDM are used for RGB and depth separately.
> >
> > As mentioned above, we do *not* use two parallel LDM's. Nonetheless, we conducted the runtime analysis as requested (also by reviewer MEmh). For training, we compared the average training time (minutes) for training one epoch, on the same hardware, batch size (4) and training dataset. For inference, we compared the time (seconds) required to infer a single image.
> >
> > | Method        | Type                | Depth | Training (mins/epoch) | Inference (sec/image) |
> > | ------------- | ------------------- | ----- | --------------------- | --------------------- |
> > | PanoDiffusion | Bi-modal LDM        | +     | 82                    | 5                     |
> > | BIPS          | GAN                 | +     | 131                   | <1                    |
> > | RePaint       | Diffusion model     | -     | 78                    | 45                    |
> > | LDM           | LDM                 | -     | 72                    | 4                     |
> > | OmniDreamer   | Transformer + VQGAN | -     | 158                   | 61                    |
> >
> > The results show that while our model is not the fastest, it remains within a reasonable range. It's also noteworthy that, compared to the original LDM framework, **our bi-modal structure achieves a significant improvement in the quality of outpainting.** **This improvement comes without a proportionate increase in resource consumption** – we observed only a modest increase of 13.8% in training time and 25% in inference time.
> >
> > [1] Zhuang C, Lu Z, Wang Y, et al. ACDNet: Adaptively combined dilated convolution for monocular panorama depth estimation. Proceedings of the AAAI Conference on Artificial Intelligence. 2022, 36(3): 3653-3661.
> >
> > [2] Shen Z, Lin C, Liao K, et al. PanoFormer: Panorama Transformer for Indoor 360∘ Depth Estimation. European Conference on Computer Vision. Cham: Springer Nature Switzerland, 2022: 195-211.

---

> > > ### Comment · Reviewer_dEjQ · 2023-11-22
> > > **Thanks for the response**
> > >
> > > Thanks for the detailed information provided by the authors.
> > >
> > > My concerns have been addressed from the response, for example, the evaluation results of using concatenation or cross-attention. The authors also provide a further comparison to some RGB-based depth estimation methods like PanoFormer and ACDNet. Therefore, I would like to increase my score to support this work.

---

### Official Review · Reviewer_MEmh · 2023-10-31

**Soundness:** 3 good
**Presentation:** 3 good
**Contribution:** 3 good
**Rating:** 8
**Confidence:** 3

**Summary:**

This paper proposed a latent diffusion model (LDM) for indoor RGB panaroma inpainting and depth map generation. During training stage, the input to the bi-modal LDM structure are RGB images and corresponding depth maps, which improve the performance of panaroma inpainting. At each stage of the denoising process in the diffusion model, the proposed alignment mechanism enhances the wraparound consistency of the results. The results indicate that the proposed PanoDiffusion excels not only by achieving a substantial performance advantage over state-of-the-art techniques in RGB-D panorama outpainting, yielding diverse and well-structured results for various mask types, but also by demonstrating the capability to generate high-quality depth panoramas.

**Strengths:**

- During inference process, there is no need to input the depth map as a guidance, only training process requires depth maps as input. It is significantly different from the previous approaches.
- This paper proposed a noval approach that involves gradually introducing camera rotations at each stage of the diffusion denoising process, resulting in a notable enhancement in achieving seamless panorama wraparound consistency.
- With the clear description of each module and step, this approach ensures that the framework is not only effective but also easily comprehensible and straightforward to follow.

**Weaknesses:**

- There is not many noval changes to the LDM framework. The authors only add a pretrained depth map encoder as a guidance.
- There are only one visual result (Figure 6) in the paper, which is not convincing. The authors should add more visual results for comparison in the supplementary materials.

**Questions:**

- How about the training and inference time compared to other frameworks? Since LDM requires a step-by-setp mechanism to generate the final results, the efficiency may be a problem.
- There are only 4 types of maskes. Each type of mask covers different portion of the whole panaroma. I recommand the authors to add a chart that describes the performance changing along the percentage of the mask. According to my understanding, with the increase of the percentage of the mask, the performance will drop.
- The mask are all continuous presented in this paper, I am curious when there are several separate masks in one panaroma, what will the performance become?

---

> ### Author Response · Authors · 2023-11-20
> **Official Comment by Authors (1/2)**
>
> We appreciate the reviewer's valuable feedback and suggestions. We would like to provide a detailed response to each comment.
>
> > Q1: There is not many noval changes to the LDM framework. The authors only add a pretrained depth map encoder as a guidance.
>
> The more obvious approach is to train an LDM with depth masks as condition, particularly during training. We avoided this, and instead chose to use a RePaint-based structure, because we want to free the model from *presuming mask distributions*. This is critical for our two-end alignment design, as masks will continually change due to rotation. As LDM or Repaint cannot individually handle this challenge properly, it provides ample motivation for us to fuse these two components, naturally combined with our novel rotational denoising. Furthermore, we advanced our model to a bi-modal structure, where depth maps prove to have significant auxiliary effect in generating well-structured results. We also consider that our discovery that the *asymmetric* condition, where having a training setting (use of depth and without masking) that is substantially *different* from the inference setting (exclusion of depth and with masking), still leads to improved performance, is an interesting novel finding.
>
> > Q2: The authors should add more visual results for comparison in the supplementary materials.
>
> Perhaps it was overlooked, but we had previously included more visual results for comparison in our **Appendix, previously Fig. 12, now 13**, including two groups of examples for each mask type. Additionally, we had also provided some examples where we project different panorama images to 3D scenes in our supplementary video.
>
> > Q3: How about the training and inference time compared to other frameworks?
>
> For training, we compared average training time (minutes) for one epoch of PanoDiffusion against baseline models on the same devices, using the same batch size 4 and the same training dataset. For inference, we compared the time (seconds) required to infer a single image.
>
> | Method        | Type                | Depth | Training (mins/epoch) | Inference (sec/image) |
> | ------------- | ------------------- | ----- | --------------------- | --------------------- |
> | PanoDiffusion | Bi-modal LDM        | +     | 82                    | 5                     |
> | BIPS          | GAN                 | +     | 131                   | <1                    |
> | RePaint       | Diffusion model     | -     | 78                    | 45                    |
> | LDM           | LDM                 | -     | 72                    | 4                     |
> | OmniDreamer   | Transformer + VQGAN | -     | 158                   | 61                    |
>
> The results show that while our model is not the fastest, it remains within a reasonable range. It's also noteworthy that, compared to the original LDM framework, **our bi-modal structure achieves a significant improvement in the quality of outpainting.** **This improvement comes without a proportionate increase in resource consumption** – we observed only a modest increase of 13.8% in training time and 25% in inference time.

---

> > ### Author Response · Authors · 2023-11-20
> > **Official Comment by Authors (2/2)**
> >
> > > Q4: I recommand the authors to add a chart that describes the performance changing along the percentage of the mask.
> >
> > We appreciate your insightful suggestion. We've conducted additional experiments with different percentages of box masks to test the performance of the model, from 10% to 90%. Specifically, **we generated box masks of different total areas in percentage**, and no depth input is provided during inference. As the results suggest, the performance of PanoDiffusion declines with the increase in the mask percentage. This trend aligns with our expectations, as larger masks inherently pose a greater challenge for panorama image outpainting.
> >
> > | Mask Percentage | FID   | sFID  | Density | Coverage |
> > | --------------- | ----- | ----- | ------- | -------- |
> > | 10%             | 6.58  | 9.80  | 1.000   | 1.000    |
> > | 30%             | 8.99  | 15.43 | 1.000   | 0.999    |
> > | 50%             | 12.63 | 20.56 | 1.000   | 0.980    |
> > | 70%             | 16.88 | 24.18 | 0.901   | 0.843    |
> > | 90%             | 22.94 | 28.04 | 0.644   | 0.661    |
> >
> > We also calculated the percentage of each type of mask used in the paper. The results show that the performance is generally consistent with the mask percentage. Except for the layout mask, where we muffle all areas except floors and ceilings, it is more challenging to precisely outpaint all the details. Consequently, even though the layout mask percentage is close to the random box mask, its FID is higher.
> >
> > | Mask Type       | Camera | NFoV  | Layout | Random Box |
> > | --------------- | ------ | ----- | ------ | ---------- |
> > | Mask Percentage | 80.6%  | 85.4% | 50.6%  | 50.5%      |
> >
> > >  Q5: The mask are all continuous presented in this paper, I am curious when there are several separate masks in one panaroma, what will the performance become?
> >
> > In our method, the design of the random box mask inherently includes 1 to 5 randomly sized bounding boxes. Consequently, **there is a probability of these masks being discontinuous**. Nonetheless, to explicitly assess our model's performance here in a more deterministic manner, we first note that for the camera mask type, the visible portions are definitely non-contiguous. So to test non-contiguous masks, we flipped the camera mask - swapping the originally visible and invisible parts to simulate this situation. This type of mask is equivalent to randomly sampling several NFoV masks and making them invisible. The quantitative results show that the model is still able to perform outpainting properly.  More qualitative results are provided in the Appendix of our paper for your reference.
> >
> > | Method        | FID  | sFID  | Density | Coverage |
> > | ------------- | ---- | ----- | ------- | -------- |
> > | PanoDiffusion | 9.38 | 12.56 | 1.000   | 0.993    |

---

> > > ### Comment · Reviewer_MEmh · 2023-11-22
> > > **Concerns Resolved**
> > >
> > > After checking the response from the authors, most of my concerns are resolved. The authors add additional experiments and analysis to demonstrate the effectiveness of the proposed method. Based on their response and changes, I would like to increase my rating.
> > >
> > > Best,

---

### Official Review · Reviewer_inST · 2023-10-31

**Soundness:** 4 excellent
**Presentation:** 4 excellent
**Contribution:** 4 excellent
**Rating:** 8
**Confidence:** 4

**Summary:**

The authors propose a way to outpaint a near field-of-view image (i.e. a normal image from a normal camera) to a panorama, which they represent as a equirectangular projected image. They use latent diffusion models to do so. Their latent model is trained on RGB-D panoramic data, but works on just RGB inputs.

One problem with outpainting panoramic images is that the left and right side of the equirectangular project image need to map to each other. The authors introduce a novel technique to do this. It works by rotating the image by 90 degrees in each denoising step in the diffusion model.

The authors compare to a variety of other techniques and show that their method is better than competing methods.

**Strengths:**

The paper is well-written and the results seem pretty good. The author’s idea to rotate the image 90 degrees in each denoising step seems very novel.

There are lots of comparisons to many other methods and the proposed method is better based on a variety of techniques.

I appreciate that the authors include the code as supplementary material!

**Weaknesses:**

It is somewhat hard to evaluate the generated panoramic images based on looking at images or fixed rotations. I would be interested in seeing the panoramic images of both the proposed method and competing method in a viewer like threejs. See https://threejs.org/examples/webgl_panorama_equirectangular.html

**Questions:**

There are many ways to represent 360 degree panorama images. The authors should clarify that using the equirectangular projection is a choice they are making.

Do the authors only rotate the camera in the horizontal direction? Or are vertical rotations allowed as well?

During inference, did the authors try 180 degree shifts instead of 90 degree shifts? Or some other rotation?

What’s the difference between Fig. 2 and Fig. 3? They seem like they are both figures about how training and inference are done, but they seem to be different. Specifically, Fig. 2b does not add noise and does not reference the circular shift, while Fig. 3b does. Is Fig. 2b incorrect?

In the supplemental video, do the shown 3D Scenes at 00:12 use the generated depth maps?

---

> ### Author Response · Authors · 2023-11-20
> **Official Comment by Authors**
>
> We appreciate the reviewer's valuable feedback and suggestions. We would like to provide a detailed response to each comment. For Q3 and Q4, much of the replies here are also included in our response to reviewer SqKp.
>
> > Q1: I would be interested in seeing the panoramic images of both the proposed method and competing method in a viewer like threejs.
>
> Thank you for your suggestion! In fact, we've already completed a demo that transforms panorama images into interactive threejs for users to check the results, and we plan to release the examples upon the acceptance of our paper. As proof, we provided some examples at 00:12 in our supplemental video for a preview, which are outpainted panoramas of our PanoDiffusion.
>
> > Q2: There are many ways to represent 360 degree panorama images. The authors should clarify that using the equirectangular projection is a choice they are making.
>
> Thank you, we have clarified the choice of using equirectangular projection in the dataset section of our revised paper.
>
> > Q3: Do the authors only rotate the camera in the horizontal direction? Or are vertical rotations allowed as well?
>
> For both training and inference stages, we only consider rotations in the horizontal direction. As panorama images typically follow equirectangular projection, the distortion increases non-uniformly towards the top and bottom poles. **Introducing angle rotations in the vertical direction would lead to substantial changes in the projection results**, and require more complex preprocessing, which will increase both training and inference costs. We have added two examples where the camera has a 90-degree vertical rotation in our Appendix 1.3 Fig.~10.  Conversely, **distortion is uniform horizontally** -- image manipulation here only involves horizontal cropping and splicing, without the need for reprojection. Since the main objective of PanoDiffusion is to generate coherent and realistic panorama images, we have focused more on the spatial wrap-around consistency in the horizontal direction only. In the future, we will consider better approaches to support wrap-around consistency in all directions.
>
> > Q4:  During inference, did the authors try 180 degree shifts instead of 90 degree shifts? Or some other rotation?
>
> During inference, we have explored the effect of different rotation angles, including 180°, 90° and 45°, on the outpainting results.  The results show that the wrap-around consistency of outpainting results is improved across all settings.  Compared to 180°, 90° leads to better consistency. However, diminishing the angle further to 45° did not lead to additional improvements. We believe this is reasonable, as the model is expected to generate coherent content when the two ends are in contact for enough denoising steps. Therefore, smaller rotation angles than 90° and longer connections do not necessarily lead to more consistent results. We have incorporated this part into the Appendix of our revised paper.
>
> | Methods \ Mask Type | Camera | NFoV   | Layout | Random Box | End    |
> | ------------------- | ------ | ------ | ------ | ---------- | ------ |
> | w/o rotation        | 125.82 | 128.33 | 128.10 | 128.19     | 132.69 |
> | 180°                | 95.11  | 96.57  | 90.93  | 85.23      | 119.60 |
> | 90°                 | 90.41  | 89.74  | 88.01  | 85.04      | 116.77 |
> | 45°                 | 90.67  | 90.25  | 87.65  | 86.50      | 112.47 |
>
> > Q5: What’s the difference between Fig. 2 and Fig. 3?
>
> Fig. 2 is intended to provide an overview of our method. We had previously deliberated on integrating the circular shifts, masking operations and iteration loop into Fig. 2, but eventually felt it became excessively cluttered. So we created Fig. 3 to illustrate those operations in more detail, while excluding them from Fig. 2.
>
> More specifically, Fig. 2 presents an overview of the entire model pipeline, **where we only show the input and output of each stage and omit the specific details of circular shift and adding noise**. This is to make our two-stage bi-modal structure clearer and to highlight the difference between the inputs we use for training and inference.
>
> For Fig. 3, we instead focus on explaining how the camera rotation mechanism is implemented.  We detailed the flow of the rotation process, **including the incorporation of noisy images with different transparency to represent different noising steps.**
>
> We have clarified this difference in Fig. 2 caption in our revised paper.
>
> > Q6: In the supplemental video, do the shown 3D Scenes at 00:12 use the generated depth maps?
>
> The 3D scenes at 00:12 do not use the generated depth maps.  We only use the outpainted panorama images of PanoDiffusion and project them onto spherical coordinates using threejs, which enabled their display in an immersive 3D format.

---

> > ### Comment · Reviewer_inST · 2023-11-22
> >
> > I am satisfied with the authors rebuttal to my concerns as well as the concerns of other reviewers and maintain my positive rating.

---

### Official Review · Reviewer_SqKp · 2023-11-01

**Soundness:** 3 good
**Presentation:** 3 good
**Contribution:** 3 good
**Rating:** 6
**Confidence:** 3

**Summary:**

This paper proposes a new diffusion based method to tackle panorama image generation task. The proposed method utilizes the latent diffusion models to outpaint the area that is not originally taken by the camera. Relying on the powerful generation capability of diffusion models, the proposed method proposes to progressively apply camera rotations during the image generation process in order to enhance the generalizability. Extensive experiments show that the proposed method is able to outperforms the baseline methods significantly.

**Strengths:**

1. This paper is generally well written with strong motivation and well-organized writing. It clearly demonstrates the problem of  current pano generation and implies the proposed method is tackling the problems stated.
2. The proposed method significantly outperforms the baseline methods on the selected benchmarks.

**Weaknesses:**

1. During the training stage, does the random angle rotation only apply to horizontal direction?
2. Depth map actually provides rich information indicating the scales of the objects in the images. In order to test the generalization capability, can you provide more results of running the proposed method on other datasets?
3. This paper claims the camera rotations as one of the contributions, please provide an ablation study on how the camera rotation actually works effectively.

**Questions:**

The authors are suggested to address the concerns raised in the weaknesses section during the rebuttal period.

---

> ### Author Response · Authors · 2023-11-20
> **Official Comment by Authors**
>
> We appreciate the reviewer's valuable feedback and suggestions. We would like to provide a detailed response to each comment.
>
> > Q1: During the training stage, does the random angle rotation only apply to horizontal direction?
>
> During the training stage, we restrict random angle rotation solely to the horizontal direction. As panorama images typically follow equirectangular projection, the distortion increases non-uniformly towards the top and bottom poles. **Introducing random angle rotation in the vertical direction would lead to substantial changes in the projection results, and require more complex preprocessing, which will increase training costs.** We have added two examples where the camera has a 90-degree vertical rotation in our Appendix 1.3 Fig.~10. Conversely, **distortion is uniform horizontally** - image manipulation here only involves horizontal cropping and splicing, without the need for reprojection.
>
> In terms of training efficacy, without providing camera viewpoints as part of the input, **adding vertical rotations does not improve model performance**. Instead, it can complicate the learning of equirectangular projection patterns, reducing overall effectiveness. Since the main objective of PanoDiffusion is to generate coherent and realistic panorama images, we focus more on data augmentation in the horizontal direction - the spatial wrap-around consistency.
>
> > Q2: In order to test the generalization capability, can you provide more results of running the proposed method on other datasets?
>
> We conducted additional tests using a set of 2305 panorama images from Matterport3D [1] dataset. These tests were performed without re-training or fine-tuning PanoDiffusion, which was trained on the Structured3D dataset.
>
> |                  | Camera Mask  |        |       |       | NFoV Mask |        |       |       |
> | ---------------- | --------------- | ------ | ----- | ----- | ------------- | ------ | ----- | ----- |
> | Method \ Metrics | FID ↓           | sFID ↓ | D ↑   | C ↑   | FID ↓         | sFID ↓ | D ↑   | C ↑   |
> | PanoDiffusion | **45.52**       | **37.34** | **0.639** | **0.622** | **49.86**     | **38.31** | **0.740** | **0.649** |
> | BIPS | 83.23 | 45.97 | 0.113 | 0.098 | 97.13 | 57.56 | 0.082 | 0.045 |
> | Inpaint Anything | 65.69 | 50.67 | 0.049 | 0.121 | 79.48 | 56.57 | 0.014 | 0.031 |
>
> Matterport3D differs from Structured3D with greater complexity and diversity, including multi-story and outdoor scenes. Despite these challenges, PanoDiffusion outperforms the baseline models and is able to outpainting reasonable panorama images in most of the cases. We have provided some qualitative results in the Appendix of our paper for your reference.
>
> > Q3: This paper claims the camera rotations as one of the contributions, please provide an ablation study on how the camera rotation actually works effectively.
>
> We have previously provided some ablation results on this. **Please see Fig. 8 and Table 3, where we compared PanoDiffusion with and without the use of rotations.**
>
> To further shed light on the reviewer's question, we **additionally explored the effect of the rotation angle, including 180°, 90° (chosen for our final result) and 45°, on the outpainting results.** The results show that the wrap-around consistency is improved across all settings. Compared to 180°, 90° leads to better consistency. However, diminishing the angle further to 45° did not lead to additional improvements. We believe this is reasonable, as the model is expected to generate coherent content when the two ends are in contact for enough denoising steps. Therefore, smaller rotation angles and longer connections do not necessarily lead to more consistent results.
>
> | Methods \ Mask Type | Camera | NFoV   | Layout | Random Box | End    |
> | ------------------- | ------ | ------ | ------ | ---------- | ------ |
> | w/o rotation        | 125.82 | 128.33 | 128.10 | 128.19     | 132.69 |
> | 180°                | 95.11  | 96.57  | 90.93  | 85.23      | 119.60 |
> | 90°                 | 90.41  | 89.74  | 88.01  | 85.04      | 116.77 |
> | 45°                 | 90.67  | 90.25  | 87.65  | 86.50      | 112.47 |
>
> [1] Chang A, Dai A, Funkhouser T, et al. Matterport3d: Learning from rgb-d data in indoor environments. arXiv preprint arXiv:1709.06158, 2017

---

### Author Response · Authors · 2023-11-20
**Response to all reviewers**

We thank all reviewers for their constructive comments. We are highly encouraged by the positive comments: *"strong motivation, well-organized writing"(Reviewer SqKp)*, *"novel idea" (Reviewer inST)*, *"effective, comprehensible and straightforward" (Reviewer MEmh)*. All reviewers acknowledge the *"convincing experiments, superior performance''* and *"well-written''*.

We provided detailed responses to each reviewer. Nonetheless, we summarize some generally raised issues here.

**Novelty of PanoDiffusion** The more obvious approach is to train an LDM with depth masks as condition, particularly during training. We avoided this, and instead chose to use a RePaint-based structure, because we want to free the model from *presuming mask distributions*. This is critical for our two-end alignment design, as masks will continually change due to rotation. While RePaint or LDM cannot individually handle this challenge properly, it provides ample motivation for us to fuse these two components, naturally combined with our rotational denoising. Furthermore, we advanced our model to a bi-modal structure, where depth maps prove to have significant auxiliary effect in generating well-structured results. We also consider that our discovery that the *asymmetric* condition, where having a training setting (use of depth and without masking) that is substantially *different* from the inference setting (exclusion of depth and with masking), still leads to improved performance, is an interesting novel finding.

**Intuition about using depth information** Our motivation for using depth maps during training is centered on two aspects:

1) In realistic scenarios, we believe it is a more accessible and reliable choice than semantic maps, which may require manual annotations, or other modalities.
2) Panoramas, compared to normal images, have an internal physical structure that the wrapped-around result should be a coherent space. Therefore, depth can be especially helpful for the model to understand the scene layouts and physical structure of the objects.

**Direction of camera-rotation** For training, we used random rotations as data augmentation, while during inference, we introduced 90° rotations at each denoising step for better spatial wrap-around consistency. In both stages, we only consider rotations in the horizontal direction. As panorama images typically follow equirectangular projection, the distortion increases non-uniformly towards the top and bottom poles. Introducing angle rotations in the vertical direction would lead to substantial changes in the projection results, and require more complex preprocessing, which will increase both training and inference costs. Conversely, *distortion is uniform horizontally* -- image manipulation here only involves horizontal cropping and splicing, without the need for reprojection. Since the main objective of PanoDiffusion is to generate coherent and realistic panorama images, we have focused more on the spatial wrap-around consistency in the horizontal direction only. In the future, we will consider better approaches to support wrap-around consistency in all directions.

**Efficiency of PanoDiffusion** From the perspective of training, PanoDiffusion can be trained within a day to achieve the results we report in our paper, faster than the GAN-based and transformer-based models for image inpainting or panorama outpainting. From the perspective of inference, although our model is not the fastest, it remains within a reasonable and acceptable range.  Furthermore, our bi-modal structure does not significantly increase the training and inference time compared to the original LDM, while achieving a notable improvement in the quality of outpainting.

As mentioned, we provided further detailed answers to each reviewer's questions, including both quantitative and qualitative (attached in our Appendix) results. Reviewers are also welcome to check our supplemental video for the threejs results.

**Summary of changes** Here is the summary of changes to our paper (changes marked in blue):

- Clarified our intentional omission of the specific details of circular shift and adding noise in Fig. 2 caption
- Clarified the choice of using equirectangular projection in Section 4.1
- Examples of vertical camera-rotation in Appendix A.1.3 Fig. 10
- Ablation study on camera-rotation Angles in Appendix A.2.3 Table 6
- Qualitative results of zero-shot test on Matterport3D dataset in Appendix A.2.6 Fig. 15
- Qualitative results of discrete mask ablation in Appendix A.2.7 Fig. 16
- Qualitative results of synthesized RGB-D panorama results in Appendix A.2.7 Fig. 17
- Runtime analysis results in  Appendix A.3 Table 7

---

### Meta-Review · Area_Chair_GLma · 2023-12-13

**Metareview:**

This paper introduces a novel latent diffusion model for generating panoramic images, enhancing the field of image generation by addressing the limitations of current methodologies. By utilizing depth information during the training stage and applying rotations in the diffusion process, the proposed method, PanoDiffusion, achieves high-quality RGB-D panorama generation. The strength of the paper lies in its ability to outperform baseline methods and produce significant improvements in panoramic image quality.

(a) Strengths of the paper

The paper's strengths, as acknowledged by Reviewers SqKp, inST, MEmh, and dEjQ, include:

- A well-articulated methodology with a strong motivation and clear writing style.

- Significant performance improvements over baseline methods, confirmed by extensive experiments.

- The introduction of camera rotations during the diffusion process as a novel data augmentation technique, enhancing wraparound consistency.

- The inclusion of code as supplementary material, enhancing the paper's reproducibility and transparency.

(b) Weaknesses of the paper

There is a noted need for additional visual results and comparisons, which are crucial for fully establishing the effectiveness of the proposed model in various scenarios (Reviewers MEmh and dEjQ). Reviewers have expressed the necessity for a more comprehensive evaluation, including the potential use of interactive viewers like three.js, to better assess the quality of the panoramic images generated (Reviewer inST). The actual contribution of depth information during inference and its necessity for the model has been questioned, suggesting that further clarification and justification are needed to understand its role within the model (Reviewers SqKp and dEjQ).

In conclusion, the paper presents a valuable contribution with the potential for substantial impact.

**Justification For Why Not Higher Score:**

Based on the feedback from reviewers, the paper's moderate improvements in the field of panoramic image generation and the need for further validation. Although the approach shows promise and outperforms baselines, it requires deeper empirical evidence across diverse conditions to reach a higher conference profile. Additionally, the paper's explanation of depth information's role in inference needs more clarity. These gaps, alongside the lack of interactive visual results, suggest that a poster session is most appropriate.

**Justification For Why Not Lower Score:**

This paper presents a well-founded approach to panoramic image generation with demonstrated improvements over existing baselines. Reviewers recognize the method's potential and the robustness of initial results. While there are calls for more extensive validation and clarity on certain aspects, these are not insurmountable flaws but areas for further development. The inclusion of depth information is an innovative step, but it needs more explanation. Moreover, the paper is praised for its clear writing and contribution to the field. Such constructive feedback indicates that the work has merit and provides a foundation for future advancements.

---

### Decision · Program_Chairs · 2024-01-16

Accept (poster)